# Taxonomy of reduction matrices for Graph Coarsening

**Antonin Joly**
CNRS, IRISA, Rennes, FRANCE
`antonin.joly@irisa.fr`

**Nicolas Keriven**
CNRS, IRISA, Rennes, FRANCE
`nicolas.keriven@cnrs.fr`

**Aline Roumy**
INRIA, Rennes, FRANCE
`aline.roumy@inria.fr`

## Abstract

Graph coarsening aims to diminish the size of a graph to lighten its memory footprint, and has numerous applications in graph signal processing and machine learning. It is usually defined using a reduction matrix and a lifting matrix, which, respectively, allows to project a graph signal from the original graph to the coarsened one and back. This results in a loss of information measured by the so-called Restricted Spectral Approximation (RSA). Most coarsening frameworks impose a fixed relationship between the reduction and lifting matrices, generally as pseudo-inverses of each other, and seek to define a coarsening that minimizes the RSA.

In this paper, we remark that the roles of these two matrices are not entirely symmetric: indeed, putting constraints on the *lifting matrix alone* ensures the existence of important objects such as the coarsened graph's adjacency matrix or Laplacian. In light of this, in this paper, we introduce a more general notion of reduction matrix, that is *not* necessarily the pseudo-inverse of the lifting matrix. We establish a taxonomy of "admissible" families of reduction matrices, discuss the different properties that they must satisfy and whether they admit a closed-form description or not. We show that, for a *fixed* coarsening represented by a fixed lifting matrix, the RSA can be *further* reduced simply by modifying the reduction matrix. We explore different examples, including some based on a constrained optimization process of the RSA. Since this criterion has also been linked to the performance of Graph Neural Networks, we also illustrate the impact of this choices on different node classification tasks on coarsened graphs.

## 1 Introduction

In recent years, several applications in data science and machine learning have produced large-scale *graph* data [18, 5]. For instance, online social networks [13] or recommender systems [36] routinely produce graphs with millions of nodes or more. To handle such massive graphs, researchers have developed general-purpose *graph reduction* methods [4], such as **graph coarsening** [27, 7] as well as specific learning techniques on these coarsened graphs [23, 19]. Graph coarsening starts to play an increasingly prominent role in machine learning applications [7].

**Graph Coarsening and Spectral guarantees** Graph coarsening consists in producing a small graph from a large graph while retaining some of its key properties. There are many ways to evaluate the quality of a coarsening, following different criteria [10, 27, 7]. The majority of these approaches aims to preserve spectral properties of the graph and its Laplacian, and have given rise to different coarsening algorithms [27, 8, 4, 22, 28]. The most widely used spectral guarantee is the so-called *Restricted Spectral Approximation* (RSA, see Sec. 2), introduced by Loukas [27]. In broad terms,

39th Conference on Neural Information Processing Systems (NeurIPS 2025).

the RSA states that the frequency content of a certain subspace of graph signals is approximately preserved when projected on the coarsened graph and then re-lifted in the original one, or intuitively, that the coarsening is well-aligned with the low-frequencies of the Laplacian. The RSA is a general-purpose criterion with many applications, from clustering to signal reconstruction [27, 23]. Recently, RSA guarantees were also used to guarantee the performances of Graph Neural Networks on the coarsened graph [23].

**Projection and lifting.** The projection and re-lifting operations are generally described by two matrices: the *reduction matrix* allows to transform graph signals from the original graph to the coarsened one, while the *lifting matrix* does the opposite. In virtually all works on graph coarsening, these two matrices are simply pseudo-inverse of each other, and both can represent the "graph coarsening" indifferently. However, in this paper we make the following remark: *their roles are not entirely symmetric*. Mathematically, as we will see, the *lifting matrix alone* has to be quite constrained for the graph coarsening to be "well-defined", with a consistent adjacency matrix and Laplacian. The reduction matrix, on the other hand, does not seem to play a role in the *definition* of graph coarsening. However, it *does* play a role in computing the RSA. Therefore, in this paper, we examine the following questions: for a *fixed* lifting matrix, 1) **What are the admissible degrees of freedom for the reduction matrix?** and 2) **Can we improve the RSA by simple modification of the reduction matrix alone?**

**Contribution.** In this paper, we thus define and then explore the admissible sets of reduction matrices over which to optimize the RSA. We introduce several interesting examples, from closed-form ones motivated by notions of optimality and memory footprint to optimization-based ones over well-defined sets with various properties. We compare these different choices, both in terms of RSA and performance when used within GNNs trained on coarsened graphs.

**Related work** Graph coarsening is derived from the multigrid-literature [33] and belongs to a broader class of methods commonly referred to as *graph reduction*. The latter includes graph sampling [17], graph sparsification [34, 1, 26], or more recently graph distillation [21, 41, 20], inspired by dataset distillation [37]. Some of the first coarsening algorithms were linked to the graph clustering community, e.g. [9] which used recursively the Graclus algorithm [10] algorithm. Linear algebra technics such as the Kron reduction were also employed [27] [12]. In [27], the author introduces the RSA, and presents a greedy algorithm that recursively merge nodes by optimizing some cost, which in turns leads to RSA guarantees. This is the approach we use in our experiments (Sec. 5). It was followed by several similar methods with the same spectral criterion [8, 4, 22, 28]. Since modern graph often includes node features, other approaches proposed to take them into account in the coarsening process, often by learning the coarsening with specific regularized loss [25, 29, 11]. On the contrary, the RSA guarantees [27] leveraged in this paper are *uniform* over a whole subspace to ensure the spectral preservation of the coarsened graph.

Closer to us, some works aim to optimize various quantities after the coarsening has been computed. For instance, GOREN [6] optimizes in a data-driven manner the edges' weights in the coarsened graph, which is quite different from focusing on reduction/lifting matrices as proposed here. Moreover, we consider the RSA, a general-purpose criterion not necessarily related to downstream tasks. The literature also includes different choices of reduction/lifting matrices, or propagation matrices in GNNs on coarsened graphs [23], but to our knowledge this paper is the first to put forth the idea of *decorrelating reduction and lifting matrices* up to a certain point, with precise mathematical definitions of the consequences.

Finally, we mention *graph pooling*, which is designed to mimic the pooling process in deep convolutional models on images and is somewhat related to graph coarsening in terms of vocabulary. One difference is that graph pooling tends to focus only on the reduction phase while graph coarsening focuses on repeated reduce-then-lift operations between the coarsened and original graphs. Although some pooling method can be computed as preprocessing such GRACLUS [10], the most well-known pooling methods are data-driven and fully differentiable (Diffpool [40], top-K pooling [14], DMoN [35]). These methods use interchangeably the reduction and the lifting matrix by choosing one as the *transposed* of the other. Usually in graph pooling, this matrix is unconstrained and is either defined heuristically or learned, while graph coarsening proposes mathematical links between the reduction and the lifting matrix. In this paper, we explore these mathematical links.

**Outline.** We start by background material on graph coarsening in Sec. 2, highlighting the roles of the lifting and reduction matrices. We emphasize the asymmetricity of their roles, along with the strong constraints put on the lifting matrix alone. Then in Sec. 3, we study sets of admissible reduction matrices, focusing on various notions of "generalized inverses". Some will be very generic, while others will admit parametrizations that are convenient for optimization. In Sec. 4, we then study several motivated examples of reduction matrices, classical or entirely novel, with an analytical closed-form expression or based on optimization procedures. We relate them to the properties defined in the section before. Finally, we compare their performance in terms of RSA or GNN performance in Sec. 5. The code is available at `https://gitlab.inria.fr/anjoly/taxonomy-coarsening-matrices`, and proofs are deferred to App. B.

## 2 Characterizing graph coarsening with the lifting matrix

**Notations** A graph $G$ with $N$ nodes is described by its weighted adjacency matrix $A \in \mathbb{R}^{N \times N}$. The combinatorial Laplacian is defined as $\mathcal{L} = \mathcal{L}(A) := D - A$, where $D = D(A) := \mathrm{diag}(A 1_N)$ is the diagonal matrix of the degrees. A matrix is said to be **binary** if all its coefficients are either 0 or 1. For a symmetric positive semi-definite (p.s.d) matrix $M$, we denote $\|x\|_M = \sqrt{x^\top M x}$ the Mahalanobis semi-norm associated with $M$.

**Coarsening** The goal of coarsening is to reduce the size of a graph $G$ with $N$ nodes to a coarsened graph $G_c$ with $n < N$ nodes. The proportion of reduction achieved is measured by the **coarsening ratio** $r = 1 - \frac{n}{N}$. The mapping from $G$ to $G_c$ is obtained by grouping set of nodes in $G$ to form supernodes in $G_c$. This mapping can be represented by a matrix $\mathcal{Q} \in \mathbb{R}^{N \times n}$, where $\mathcal{Q}_{ik} > 0$ means that node $i$ from $G$ has been mapped onto the supernode $k$ of $G_c$. To represent a true mapping from the original nodes to the supernodes, the matrix $\mathcal{Q}$ needs to be **well-partitioned**.

**Definition 1** (Well-partitioned $\mathcal{Q}$ matrix). *$\mathcal{Q}$ is said to be well-partitioned if it has exactly **one** non-zero coefficient per row.*

One natural way [27] to define the adjacency matrix of the coarsened graph is then to take

$$A_c = \mathcal{Q}^\top A \mathcal{Q} \tag{1}$$

In addition to be well-partitioned, it is also natural to impose that $\mathcal{Q}$ is binary, as shown by [27, Prop.7] in the following Lemma.

**Lemma 1** ([27]). *Let $\mathcal{Q}$ be a well-partitioned matrix. The two following properties are equivalent: (i) $\mathcal{Q}$ is proportional to a binary matrix; (ii) for all $A$, we have $\mathcal{L}_c := \mathcal{L}(A_c) = \mathcal{Q}^\top \mathcal{L}(A) \mathcal{Q}$.*

In other words, the Laplacian of the coarsened graph can be defined by the same equation as (1). In light of this, matrices $\mathcal{Q}$ in this paper will always be well-partitioned and binary. This results in a particularly interpretable $A_c$: a weighted edge between two supernodes has a value equal to the sum of the weights of all the edges between the two groups of original nodes.

**Lifting, Reduction, and spectral quality measure for Graph Coarsening** The quality of a coarsening can be assessed from a signal processing point of view [27]. Indeed, one way to interpret $\mathcal{Q}$ is that it can be used to "lift" a signal $y \in \mathbb{R}^n$ from the coarsened graph to the original one, as $x = \mathcal{Q}y$, and is thus called the **lifting matrix** in the literature. Its counterpart is a **reduction matrix** $P \in \mathbb{R}^{n \times N}$ that reduces a signal from $G$ to $G_c$. More formally, let $x \in \mathbb{R}^N$ be a signal over the nodes of $G$. The coarsened signal $x_c \in \mathbb{R}^n$ and the re-lifted signal $\tilde{x} \in \mathbb{R}^N$ are defined by

$$x_c = Px, \qquad \tilde{x} = \mathcal{Q}x_c = \Pi x \tag{2}$$

where $\Pi = \mathcal{Q}P$. To measure the quality of the coarsening, a popular criterion introduced by Loukas [27], is then the *Restricted Spectral Approximation* (RSA), which measures the loss of information from $x$ to $\tilde{x}$. Since $\Pi$ is at most of rank $n < N$, only a subspace $\mathcal{R}$ of $\mathbb{R}^N$ may be preserved. This leads to the definition of the RSA constant below.

**Definition 2** (Restricted Spectral Approximation constant). *Consider a subspace $\mathcal{R} \subset \mathbb{R}^N$, a Laplacian $\mathcal{L}$, a lifting matrix $\mathcal{Q}$, a reduction matrix $P$. The RSA constant $\epsilon_{\mathcal{L},\mathcal{Q},P,\mathcal{R}}$ is defined as*

$$\epsilon_{\mathcal{L},\mathcal{Q},P,\mathcal{R}} = \sup_{x \in \mathcal{R}, \|x\|_{\mathcal{L}}=1} \|x - \mathcal{Q}Px\|_{\mathcal{L}} \tag{3}$$

Classically, the preserved subspace $\mathcal{R}$ is spanned by the eigenvectors of the first eigenvalues of $\mathcal{L}$. In this case, the RSA constant can be used to bound the deviation between the spectrums of $\mathcal{L}$ and $\mathcal{L}_c$. When $\mathcal{R}$ is an eigen-subspace of $\mathcal{L}$, the RSA has an explicit expression [27]: $\epsilon_{\mathcal{L},\mathcal{Q},P,\mathcal{R}} = \|\mathcal{L}^{1/2}(I_N - P\mathcal{Q})VV^T\mathcal{L}^{+1/2}\|_2$ where $V$ is an orthogonal basis of $\mathcal{R}$ and $\|\cdot\|_2$ is the spectral norm. Note that this expression is convex in $P$, which will be useful for optimization. This definition slightly differs from [27], as it disentangles the roles of the lifting matrix $\mathcal{Q}$ and the reduction matrix $P$, while in [27], $\mathcal{Q}$ was fixed as the Moore Penrose inverse of $P$.

$\mathcal{Q}$ **is more constrained than** $P$    One might have noticed that $P$ and $\mathcal{Q}$ do not exactly play symmetric roles. Indeed, and this is the first key message of the paper: as shown by (1) and Lem. 1, **the matrix $\mathcal{Q}$ alone fully characterizes the graph coarsening**, and *the matrix $\mathcal{Q}$ alone must respect strong constraints* (it must be well-partitioned and binary). Technically, the reduction matrix $P$ does not play any role in the definition of $A_c$ or $\mathcal{L}_c$.

However, $P$ still plays a role in the computation of the RSA constant. As mentioned in the introduction, in virtually every formulations of graph coarsening, $P$ is taken as the Moore-Penrose pseudo-inverse $P = \mathcal{Q}^+$. For a well-partitioned, binary $\mathcal{Q}$, this matrix has the same support as $\mathcal{Q}^\top$, with rows that contain only coefficients $1/n_k$ where $n_k$ is the size of the $k$th supernode. In this paper, we challenge this choice and argue that *there is no real reason for it*. As we will see, there is a relative degree of freedom in designing $P$, and this is our second main message: *for a fixed $\mathcal{Q}$, the matrix $P$ can be optimized to improve the RSA constant*. Of course, this optimization may still satisfy important constraints in terms of interpretability, feasibility, memory footprint, or just simplicity, and this forms the main questions mentioned in the introduction: given a well-partitioned and binary lifting matrix, what are the "valid" reduction matrices? Is there a more "optimal" choice to minimize the RSA?

**Normalized Laplacian matrices**    Before moving on to the next section, we adapt the previous discussion to a broader notion of *"normalized" Laplacian*, that we call $\Delta$-Laplacian, defined as
$$L = L(A) = \Delta\mathcal{L}\Delta \tag{4}$$
where $\mathcal{L} = \mathcal{L}(A)$ is the combinatorial Laplacian, and $\Delta = \Delta(A) \in \mathbb{R}^{N\times N}$ a strictly positive diagonal matrix that depends on the adjacency matrix. This $\Delta$-Laplacian thus encompasses the combinatorial Laplacian when $\Delta = I_N$, or the classical normalized Laplacian when $\Delta = D^{-1/2}$. Another interesting example is the self-loop normalized Laplacian $\Delta = (D + I_N)^{-1/2}$, which is such that $L = I_N - S$ where $S$ is the propagation matrix of the classical Graph convolution network defined by Kipf [24].

To extend the definition of the RSA constant (3) to the case of the generalized $\Delta$-Laplacian, it is first necessary to ensure that the norms used in the original and coarsened graphs, $G$ and $G_c$, are comparable [27, Corollary 12]. This requires establishing the consistency of the $\Delta$-Laplacian matrices of $G$ and $G_c$. This is shown with the following lemma, provided that, starting from any binary well-partitioned lifting matrix $\mathcal{Q} \in \mathbb{R}^{N\times n}$, a generalized lifting matrix is constructed as
$$Q = Q(A, \mathcal{Q}) := \Delta^{-1}\mathcal{Q}\Delta_c. \tag{5}$$
where $\Delta_c = \Delta(A_c)$ with $A_c$ defined in (1). Note that $Q$ is also well-partitioned when $\mathcal{Q}$ is well-partitioned, however it is generally not binary. Instead, the constraint still lies on $\mathcal{Q}$, as shown below.

**Lemma 2** (Consistency, adaptation [27]). *Let $\mathcal{Q}$ be a well-partitioned lifting matrix. The two following properties are equivalent:*
   *a) $\mathcal{Q}$ is proportional to a binary matrix.*
   *b) For all adjacency matrices $A$, we have $L(A_c) = Q(A, \mathcal{Q})^\top L(A)Q(A, \mathcal{Q})$, where we recall that $L$ is defined in (4) and $Q$ in (5).*

Hence, the normalized Laplacian of the coarsened graph can again be directly deduced from the normalized Laplacian of the original graph when adopting the generalized lifting matrix $Q = Q(A, \mathcal{Q})$ with a well-partitioned and binary $\mathcal{Q}$. This consistency then enables the definition of a generalized RSA constant
$$\epsilon_{L,Q,P,\mathcal{R}} = \sup_{x\in\mathcal{R}, \|x\|_L=1} \|x - QPx\|_L \tag{6}$$
where, again, we emphasize that $L$ is defined in (4) and $Q$ in (5). The main question that we will examine in the rest of the paper remains: for a fixed $\mathcal{Q}$ that is well-partitioned and binary, what are the "valid" reduction matrices $P$, and is there a more "optimal" choice to minimize $\epsilon_{L,Q,P,\mathcal{R}}$?

# 3 Expanding the space of reduction matrices

In this section, we examine the first of the above questions: what are the "valid" reduction matrices $P$? One might be tempted to simply minimize the RSA equation (6) over all $P$ matrices, however this discards important properties of graph coarsening that we want to preserve. To address this, we introduce several ensembles of admissible $P$ matrices and derive key properties that will enable us, in Sec. 4, to optimize $P$ and to compare several examples.

In the beginning of this section, $Q$ will indicate any well-partitioned matrix. We note that, unlike all the matrices considered in the literature, *some matrices $P$ with a support different from $Q^\top$ will be acceptable*. Intuitively, $Q$ defines the "true" mapping between the nodes and the supernodes by enforcing the well-partitioned aspect, while $P$ will be allowed to relax this constraint. We begin by examining the largest such ensemble, denoted $E_1$.

$E_1$**: $P$ such that $\Pi$ is a projection.** A first minimal property in graph coarsening is that *applying successively coarsen and lift procedures does not degrade further the signal*, or in other words, the coarsen and lift operator $\Pi = QP$ is a projection $\Pi^2 = \Pi$.

The classical choice, where $P$ is the Moore Penrose inverse of $Q$, satisfies the projection property – $\Pi$ is even an *orthogonal* projector in this case. However it is only an example among a bigger set of reduction matrices which satisfies $\Pi^2 = \Pi$. To characterize this larger set, we first recall the notion of *generalized inverse*.

**Definition 3** (Generalized Inverse). *Let $A \in \mathbb{K}^{m \times n}$. We consider a matrix $B \in \mathbb{R}^{n \times m}$ which can satisfy the following conditions:*

  *(i)  $B \in A^g$   with   $A^g := \{M \mid AMA = A\}$*
  *(ii)  $A \in B^g$   i.e   $BAB = B$*
  *(iii)  $AB$ is Hermitian: $(AB)^* = AB$.*
  *(iv)  $BA$ is Hermitian: $(BA)^* = BA$.*

*The matrix B is said to be:*
  - ***generalized inverse*** *of A when it satisfies (i);*
  - ***generalized reflexive inverse*** *of A if it satisfies simultaneously (i) and (ii);*
  - ***Moore Penrose inverse*** *of A if it satisfies (i), (ii), (iii) and (iv).*

**Remark 1** (uniqueness). *The Moore Penrose inverse is **unique** while there may exist **infinitely** many "generalized inverses" [31].*

We now propose an alternative characterization of the $E_1$ ensemble.

**Lemma 3** (Generalized Inverse and $\Pi$ projection). *For a well-partitioned lifting matrix $Q$:*

$$\Pi^2 = \Pi \quad \Longleftrightarrow \quad Q \in P^g$$

The proof can be found in App. B.4. Lem. 3 means that, *assuming that $Q$ is well-partitioned*, $E_1 = \{P \mid \Pi \text{ projection}\} = \{P \mid Q \in P^g\}$. Note that the hypothesis on $Q$ is important here: this lemma is not true in general, and only valid because $Q$ is particularly simple.

Figure 1: Ensembles of admissible reduction matrices. $E_1$ includes all $P$ satisfying only the projection constraint for $PQ = \Pi$. $E_2$ contains all generalized inverses of $Q$, and is shown to be a subset of $E_1$ in Lem. 4. $E_3$ restricts $E_2$ to $P$ matrices sharing the same support as $Q^T$.

In general, the set $E_1$ does not admit further description, and it seems difficult to optimize over with algorithms such as projected gradient descent. We see next that a relatively minor additional constraints considerably simplifies the situation.

$E_2$**: $P$ generalized inverse of $Q$.** We now consider an ensemble characterized in a reverse manner of $E_1$. The rationale is that this set admits a closed form characterization, which allows for an easy implementation of optimization algorithms over it.

**Lemma 4** (Generalized reflexive inverse). *For a well-partitioned lifting matrix $Q$ and a reduction matrix $P$ such that $Q \in P^g$, we have the following equivalence:*

$$\text{rank}(P) = n \quad \Longleftrightarrow \quad P \in Q^g$$

*Conversely, $P \in Q^g$ implies $Q \in P^g$ and $\text{rank}(P) = n$, such that $Q^g \subset E_1$.*

Again, this proof relates specifically to well-partitioned matrices $Q$. We thus define $E_2 := Q^g$, and the lemma shows that $E_2 \subset E_1$ (Fig. 1). The inclusion is often strict, as there generally exists $P$ such that $Q \in P^g$ but such that $\text{rank}(P) < n$. The proof of this lemma can be found in App. B.5.

**Remark 2** (Generalized inverse $\Rightarrow$ reflexive). *In Lem. 4 we show that, for well-partitioned $Q$, generalized inverses of $Q$ are automatically* reflexive *generalized inverses. Of course, this is not true in general, here this is again due to the fact that $Q$ is well-partitioned.*

As hinted above, the set $E_2$ is far easier to describe than $E_1$.

**Lemma 5** (Characterization of generalized reflexive inverses of $Q$). *Let $Q \in \mathbb{R}^{N \times n}$ be a well-partitioned lifting matrix. All the reflexive generalized inverses of $Q$ can be characterized as:*

$$E_2 = Q^g = \{Q^+ + M\left(I_N - QQ^+\right) \mid M \in \mathbb{R}^{n \times N}\}$$

*with $Q^+$ the Moore Penrose inverse of $Q$.*

These two lemmas are important because they provide a way to optimize $P$. Indeed, by the converse of Lem. 4, it is shown that $E_2$ (being a subset of $E_1$) contains admissible matrices. Moreover, Lem. 5 (proven in App. B.6) offers a convenient characterization of $E_2$ through the matrix $M$.

$E_3$**: $P$ generalized reflexive inverse with same support**     Up until now, matrices in $E_1$ and $E_2$ have no reason to have the same support as $Q^+$, unlike for instance the Moore-Penrose inverse. Worse, matrices in $E_2$ may be very dense, which might hinder computation time and increase memory usage. Therefore, we consider sparser $P$, and add the constraint that $P$ has the same support as $Q^\top$, while still being a reflexive inverse. By construction, $E_3 \subset E_2$, as shown in Fig. 1. Moreover, $E_3$ is not empty, as it contains at least the Moore-Penrose inverse (see next section).

**Lemma 6** (Generalized reflexive inverse with same support). *Let $Q = Q(A, \mathcal{Q}) \in \mathbb{R}^{N \times n}$ be generalized lifting matrix with $\mathcal{Q}$ well-partitioned and binary. The set of reflexive generalized inverse of $Q$ with the same support as $Q^\top$ is defined as :*

$$E_3 = \left\{ P \in \mathbb{R}^{n \times N} \;\middle|\; \begin{cases} supp(P) = supp(Q^\top) \\ \sum_{k=1}^{N} \frac{P_{ik}}{\Delta(k)} = \frac{1}{\Delta_c(i)} & \forall i \in [1, n] \end{cases} \right\}$$

Note that, while Lem. 3, 4 and 5 were valid for *any* well-partitioned matrix $Q$, here we specifically examine $Q = Q(A, \mathcal{Q})$ with $\mathcal{Q}$ well-partitioned *and* binary. Moreover, all matrices with non-zero coefficients on the support of $Q^\top$ that are also in $E_1$ are in $E_2$ and $E_3$, as their rank is equal to $n$. To provide a better intuition about Lem. 6, consider the case of the combinatorial Laplacian ($\Delta = I_N$) where $Q = \mathcal{Q}$; this lemma reduces to $\sum_{k=1}^{N} P_{ik} = 1$, which appears to be a natural condition already used in [3].

The lemma is proved in App. B.7. Note that optimizing over $E_3$ is particularly light compared to the previous dense examples, as it requires optimizing only over the $N$ non-zero coefficients located on the support of $Q^\top$. Projected gradient descent can be implemented with a simple renormalization of the rows at each iteration.

## 4    From classical to novel reduction matrices: a comparative study

Now that we have proposed a taxonomy of the valid reduction matrices $P$, we investigate the second question raised in Sec. 2: what are good examples of reduction matrices? In all this section, we consider $Q = Q(A, \mathcal{Q})$ with $\mathcal{Q}$ well-partitioned and binary. As outlined in Sec. 2, $L_c = Q^\top L Q$ is then the $\Delta$-Laplacian of the coarsened graph. We will start with three examples with closed-form analytic expression, then outline a possible optimization framework, emphasizing that it is only one choice among many possible. It is worth noting that the transposed matrix $P = Q^\top$, commonly used in graph pooling, does not belong to $E_1$, and is therefore not considered in our analysis.

**Moore-Penrose Reduction.**    The most common choice for $P$ in the literature [27, 11, 25], is simply the Moore Penrose inverse of $Q$ (the derivation is in App. B.2):

$$P_{MP} := Q^+ = (Q^\top Q)^{-1} Q^\top \tag{7}$$

By Def. 3, $P_{MP} \in E_2$ and has the same support as $Q^\top$ since $Q^\top Q$ is diagonal when $\mathcal{Q}$ is well-partitioned. Thus, $P_{MP} \in E_3$.

Interestingly, $P_{MP}$ is the solution of the following optimization problem, which we note is formulated over *all* matrices $P \in \mathbb{R}^{n \times N}$ without any constraint:

$$\arg\min_{P} \sup_{x \in \mathbb{R}^N, \|x\|_2 = 1} \|x - QPx\|_2. \tag{8}$$

This problem looks suspiciously similar to the RSA (6), but differs in several key aspects. Namely, the signals lives in $\mathbb{R}^N$ not in $\mathcal{R}$, the Mahalanobis norm $\|\cdot\|_L$ in (6) is replaced by the Euclidean $l_2$ norm. This suggests that $P_{MP}$ is still "optimal" from a certain point of view, but for a different (much simpler) problem than the RSA. Below, we will formulate an optimal matrix for a problem closer to the RSA, but first examine another potential choice in $E_3$.

**Loukas Reduction [27], aka iterative coarsening.** Many coarsening algorithms construct the lifting matrix iteratively, as a product of individual coarsening $Q = Q_1 \ldots Q_c$. In [27], Loukas implements such an algorithm, and chooses the reduction matrix as the product of the Moore-Penrose inverses of each lifting matrix

$$P_{Loukas} = Q_c^+ \ldots Q_1^+ \tag{9}$$

Note that this is *not* equal to the Moore-Penrose inverse of $Q$ in general. However, and as shown in App. B.3, this results in a matrix $P_{Loukas} \in E_3$, (see Fig. 1).

**Rao and Mitra inspired reduction** As we have seen above, the Moore-Penrose inverse can be interpreted as the solution of an optimization problem (8), that differs from the RSA (6) in two key aspects: the subspace $\mathcal{R}$ and the Mahalanobis norm. We now consider the following problem, where we reintroduce the latter:

$$\arg\min_{P} \sup_{x \in \mathbb{R}^N, \|x\|_L = 1} \|x - QPx\|_L \tag{10}$$

Rao and Mitra show that (10) admits a unique solution under some hypotheses. In our case, several simplification over their original result happen, and we obtain the following: if $L$ **and** $L_c$ **are positive definite**, then the optimal solution is $P = L_c^{-1} Q^\top L$. This solution does not technically apply when $L$ and $L_c$ are $\Delta$-Laplacians since they are not invertible, but inspired by this, we propose the following reduction matrix:

$$P_{opt} = L_c^+ Q^\top L \tag{11}$$

Note that, even though (10) is again an optimization problem with no constraints on $P$, it is easy to check that $P_{opt} \in E_1$. However $P_{opt} \notin E_2$ as it is not full rank (see Fig. 1). Hopefully, $P_{opt}$ should lead to a better RSA constant than $P_{MP}$, even though $\mathcal{R}$ is still absent from (10). However, its main drawback is that it is dense in general.

**Optimization based Reduction** We now turn to the true RSA minimization of (6). To our knowledge, it does not have a simple solution such as $P_{MP}$ for (8) or $P_{opt}$ for (10), so that we need to implement an iterative optimization algorithm. As discussed in the previous section, it is particularly convenient to minimize over the set $E_2$, thanks to the characterisation of $E_2$ with Lem. 5. The minimization can then be written as:

$$P_g^* = \Phi_Q(M^*) \quad \text{with} \quad M^* = \arg\min_{M \in \mathbb{R}^{n \times N}} \sup_{x \in \mathcal{R}, \|x\|_L = 1} \|x - Q\Phi_Q(M)x\|_L \tag{12}$$

with $\Phi_Q(M) = Q^+ + M(I_N - QQ^+)$. Again, solutions $P_g^*$ are usually dense. A potential remedy is to add a sparsity constraint on $P$, which leads to the following problem:

$$P_{g,l_1}^* = \Phi_Q(M_{l_1}^*) \quad \text{with} \quad M_{l_1}^* = \arg\min_{M} \sup_{x \in \mathcal{R}, \|x\|_L = 1} \|x - Q\Phi_Q(M)x\|_L + \lambda \|\Phi_Q(M)\|_1 \tag{13}$$

As mentioned before, when $\mathcal{R}$ is a subspace of $L$, the RSA has a close form expression that is convex in $P$. This results in optimization problem that are convex in $M$ in these cases. In our experiments, we treat the $l_1$ penalty simply with gradient descent combined with a final thresholding operation (for simplicity we leave aside more complex optimization procedures with e.g. proximal operators).

Finally, we also mentioned that optimization over $E_3$ was also particularly simple: a simple renormalization is sufficient for projected gradient descent. It has the advantage of being always sparse, as it respects the support of $Q^\top$ (of size $N$). This leads to the following problem:

$$P_{Q^\top}^* = \arg \min_{P \in E_3} \sup_{x \in \mathcal{R}, \|x\|_L = 1} \|x - QPx\|_L \tag{14}$$

This is again convex in $P$, as the constraint $P \in E_3$ is linear.

## 5   Experiments

In Sec. 4, we have proposed several examples of reduction matrices $P$ that aimed to minimize the RSA with various constraints. In this section, we evaluate numerically the performance of these examples, both in terms of RSA constant and used within GNNs.

**Setup:   (i) Graph.** We consider the two classical medium-scale graphs Cora [30], and Citeseer [15], and use the public split from [39] for training the GNNs. We restrict ourselves to medium-scale graphs because handling larger graphs presents challenges. Indeed, in the optimizations we propose, the RSA requires computing the square root of the original Laplacian, which is not sparse in general and, for graphs like Reddit [16], cannot be stored on modern GPUs . Furthermore, we only consider the largest connected component since connected graphs are better suited for coarsening (see details in App. C). This may induce some slight difference with other reported results on these datasets.

**(ii) Laplacian and preserved space.** We choose two different Laplacians: the combinatorial Laplacian $\mathcal{L}$ and the self-loop normalized Laplacian $L = \Delta \mathcal{L} \Delta$ with $\Delta = (\mathrm{diag}(A1_N) + 1)^{-1/2}$. The motivation for this Laplacian comes from the fact that it is related to the propagation matrix $S = \hat{D}^{-1/2}(A + I_N)\hat{D}^{-1/2}$ (via $L = I_N - S$) commonly used in GNNs [24, 23]. For the RSA, the preserved space $\mathcal{R}$ is chosen as the $K = 100$ first eigenvectors of $\mathcal{L}$ and $L$.

**(iii) Lifting matrix $Q$.** The method that has introduced the notion of RSA [27] is particularly relevant for computing $Q$, as we share the same objective of reducing the RSA constant. However, it requires adaptation because this method minimizes the RSA constant for the combinatorial Laplacian, and yields a binary well-partitioned matrix lifting matrix $\mathcal{Q}$. We generalize this method to minimize the RSA constant for a $\Delta$-Laplacian, resulting in a lifting matrix $Q$. It is worth noting that some of the assumptions made in [27] no longer hold in the $\Delta$-Laplacian setting. Nevertheless, we observe that this generalized construction still achieves good RSA constants. The details of the generalized algorithm are provided in App. F.

**RSA minimization.**   The RSA constants achieved by the reduction matrices introduced in Sec. 4 are shown in Fig. 2a for the combinatorial Laplacian $\mathcal{L}$ and in Fig. 2b for the self-loop normalized Laplacian $L$. The coarsening ratios range from 0.05 to 0.85. As expected, the two proposed methods $P_{opt}$ and $P_g^*$ achieve better RSA constants than the usual $P_{MP}$ and $P_{Loukas}$. Moreover, the performance gap increases with higher reduction ratios. Another interesting observation is that the optimization methods $P_{g,l_1}^*$ and $P_{Q^\top}^*$ that incorporate sparsity constraints also perform very well. At low reduction ratios, their performance is nearly indistinguishable from their unconstrained counterparts $P_g^*$. This is particularly surprising given that the reduction in the number of non-zero coefficients is around $99.8\%$ , regardless of the coarsening ratios (see App. G.2 for a detailed analysis on the sparsity of the $P$ matrices and App. E for their computational hyperparameters). We also observe that the two approaches $P_{g,l_1}^*$ and $P_{Q^\top}^*$ yield similar performance. This may be due to the strong regularization parameter used in the sparsity penalty in $P_{g,l_1}^*$; a different setting could yield a more balanced trade-off for $P_{g,l_1}^*$ between the performance of $P_g^*$ and $P_{Q^\top}^*$.

**GNN application**   Inspired by the paper [23] which link the training of Graph Neural Networks (GNN) and the RSA : we have trained for three different coarsening ratio ( $r = \{0.3, 0.5, 0.7\}$) a convolutional GNN [24] and a Simplified convolution network (SGC [38]) on Cora [30] and Citeseer [15]. Each training is averaged on 10 random split, using the same experimental setting as in [23], and the hyperparameters are provided in App. G.1. For the training on coarsened graphs, we used the propagation matrix $S_c^{MP} = PSQ$ from the paper [23], where $S - L$ and $L$ is the self-loop normalized Laplacian. This matrix depends on both the reduction matrix $P$ and the lifting matrix $Q$.

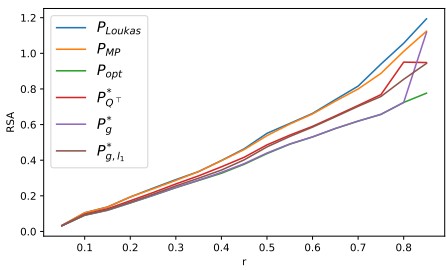
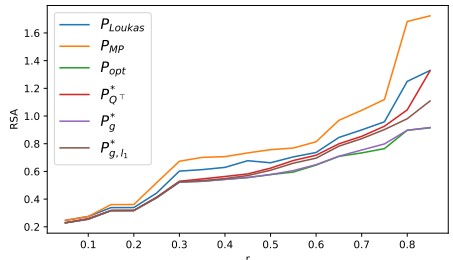

(a) Cora graph, combinatorial Laplacian $\mathcal{L}$      (b) Cora graph, self-loop normalized Laplacian $L$

Figure 2: RSA for different reduction matrices

The results reported in Tab. 1 show slightly better performances on Cora for high coarsening ratio and RSA-optimized reduction matrices such as $P_g^*$, $P_{g,l_1}^*$ and $P_{Q^\top}^*$. The results on CiteSeer are less pronounced, which may be due to its higher level of heterophily compared to Cora: indeed, since spectral coarsening is designed to preserved the low frequencies of the Laplacian and [23] the RSA is more relevant when homophily is high. Lastly, we note that $P_{opt}$ has the best RSA but poor GNN performance. We might explain this by its *density* which implies a propagation with $S_c^{MP}$ that is similar to a complete graph. The sparsity of each matrix can be found in App. G.2. Therefore, the RSA only relatively translates to a better GNN accuracy, mitigating the theoretical results of [23].

Table 1: Accuracy in % for node classification with SGC and GCNconv on different coarsening ratio

| SGC | Cora | | | Citeseer | | |
|---|---|---|---|---|---|---|
| $r$ | 0.3 | 0.5 | 0.7 | 0.3 | 0.5 | 0.7 |
| $P_{Loukas}$ | $80.5 \pm 0.0$ | $79.7 \pm 0.0$ | $76.8 \pm 0.0$ | $72.6 \pm 0.3$ | $71.7 \pm 0.1$ | $\mathbf{69.7} \pm 0.7$ |
| $P_{MP}$ | $80.5 \pm 0.0$ | $\mathbf{80.1} \pm 0.0$ | $77.7 \pm 0.0$ | $72.8 \pm 0.5$ | $72.7 \pm 0.0$ | $69.5 \pm 0.3$ |
| $P_{opt}$ | $77.1 \pm 0.6$ | $75.9 \pm 0.1$ | $73.8 \pm 0.3$ | $70.9 \pm 0.2$ | $70.2 \pm 0.1$ | $67.3 \pm 0.4$ |
| $P_{Q^\top}^*$ | $80.3 \pm 0.0$ | $80.0 \pm 0.1$ | $77.2 \pm 0.0$ | $72.7 \pm 0.3$ | $72.6 \pm 0.5$ | $67.6 \pm 0.2$ |
| $P_g^*$ | $\mathbf{80.7} \pm 0.0$ | $80.0 \pm 0.0$ | $77.6 \pm 0.0$ | $72.6 \pm 0.2$ | $\mathbf{72.7} \pm 0.0$ | $68.6 \pm 0.4$ |
| $P_{g,l_1}^*$ | $80.4 \pm 0.0$ | $79.2 \pm 0.0$ | $\mathbf{78.3} \pm 0.0$ | $\mathbf{73.0} \pm 0.0$ | $71.2 \pm 0.1$ | $69.2 \pm 0.4$ |
| Full Graph | | $81.0 \pm 0.1$ | | | $71.6 \pm 0.1$ | |
| GCN | Cora | | | Citeseer | | |
| $r$ | 0.3 | 0.5 | 0.7 | 0.3 | 0.5 | 0.7 |
| $P_{Loukas}$ | $\mathbf{80.6} \pm 0.8$ | $80.5 \pm 1.0$ | $78.1 \pm 1.4$ | $71.0 \pm 1.6$ | $72.2 \pm 0.6$ | $70.4 \pm 0.8$ |
| $P_{MP}$ | $80.4 \pm 1.0$ | $80.7 \pm 0.9$ | $78.6 \pm 0.9$ | $70.8 \pm 1.9$ | $72.1 \pm 1.0$ | $\mathbf{71.0} \pm 1.0$ |
| $P_{opt}$ | $73.7 \pm 1.5$ | $63.3 \pm 1.4$ | $55.11 \pm 2.4$ | $64.6 \pm 0.7$ | $50.4 \pm 1.6$ | $42.6 \pm 4.0$ |
| $P_{Q^\top}^*$ | $80.5 \pm 0.9$ | $80.9 \pm 0.6$ | $78.0 \pm 0.9$ | $\mathbf{71.1} \pm 1.5$ | $\mathbf{72.3} \pm 0.7$ | $70.0 \pm 0.9$ |
| $P_g^*$ | $\mathbf{80.6} \pm 1.1$ | $\mathbf{81.3} \pm 0.6$ | $\mathbf{78.7} \pm 0.9$ | $\mathbf{71.1} \pm 1.7$ | $72.1 \pm 1.2$ | $69.6 \pm 1.0$ |
| $P_{g,l_1}^*$ | $80.4 \pm 0.9$ | $80.0 \pm 0.9$ | $78.2 \pm 0.7$ | $70.2 \pm 1.8$ | $66.8 \pm 1.1$ | $66.7 \pm 1.2$ |
| Full Graph | | $81.3 \pm 0.8$ | | | $70.9 \pm 1.4$ | |

## 6 Conclusion

In this paper, we highlighted the crucial role of the lifting matrix $Q$ in graph coarsening. Surprisingly, we found that we can take advantage of the degree of freedom over the reduction matrix to obtain better spectral guarantees, without changing the coarsening itself or the lifting matrix. We have defined various sets of "admissible" reduction matrices with different properties, from the very generic property of simply obtaining a projection $\Pi$, to convenient parametrization with reflexive generalized inverses, and support constraints. We then showed that the classical choices of reduction matrices can be outperformed, both by well-motivated novel examples with analytic expressions, or by matrices resulting from various optimization processes with sparsity or support constraints. Even if previous works linked the performances of GNNs trained on coarsened graph with the RSA, we empirically showed that the benefits of improving the RSA were somewhat marginal, although

visible for high coarsening ratio and homophilous graphs. This suggests that other factors, such the sparsity of the reduction matrix, could also play in GNNs training.

In this work, we have selected the RSA as a general-purpose score to optimize, however our theoretical characterization of the admissible sets of reduction matrices does not particularly rely on it. We thus believe that considering more complex scoring function to take into account graphs heterophily or node features while being scalable to larger graphs is a major path for future works. Notably, our optimization framework is agnostic to the specific choice of scoring function and coarsening algorithm and directly applies to these extensions.

## Acknowledgments and Disclosure of Funding

The authors acknowledge the fundings of France 2030, PEPR IA, ANR-23-PEIA-0008 and European Union ERC-2024-STG-101163069 MALAGA.

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

# A  Useful property and definitions

**Definition 4** (Left inverse). *If the matrix $A$ has dimensions $m \times n$ and $\mathrm{rank}(A) = n$, then there exists an $n \times m$ matrix $A_L^{-1}$, called the left inverse of A, such that:*

$$A_L^{-1} A = I_n$$

*where $I_n$ is the $n \times n$ identity matrix.*

**Definition 5** (Right inverse). *If the matrix $A$ has dimensions $m \times n$ and $\mathrm{rank}(A) = m$, then there exists an $n \times m$ matrix $A_R^{-1}$, called the right inverse of A, such that:*

$$A A_R^{-1} = I_m$$

*where $I_m$ is the $m \times m$ identity matrix.*

**Lemma 7** (rank of well-partitioned matrices). *For a well-partitioned lifting matrix $Q \in \mathbb{R}^{N \times n}$,*

$$\mathrm{rank}(Q) = n \tag{15}$$

*Proof.* For a well-partitioned lifting matrix $Q \in \mathbb{R}^{N \times n}$, there is only one non zero value per row. Consequently all the columns are independent and the rank of the rank of this matrix is equal to $n$. □

**Remark 3.** *Similarly a reduction matrix $P \in \mathbb{R}^{n \times N}$, which has the same support as $Q^\top$ with $Q$ a well-partitioned lifting matrix is also of rank $n$.*

**Lemma 8** ($QQ^\top$ is diagonal). *Let $Q$ be a well-partitioned lifting matrix. For all matrix $P \in \mathbb{R}^{n \times N}$ with the same support as $Q^\top$, $PQ$ is diagonal.*

*Proof.* Let $P$ a reduction matrix with the same support as $Q^\top$. Let's show that ($Q^\top Q$ is diagonal) :

For $i, j \in \{1 \ldots N\}^2$ and $i \neq j$:

$$(Q^\top Q)_{ij} = \sum_{k=1}^{N} Q_{kj} Q_{ki}$$
$$= 0$$

Indeed if one term is different from $0$, it means that two nodes $i$ and $j$ of $G_c$ have been expanded to a same node $k$ of $G$ which contradicts the well-partitioned definition.

Thus $Q^\top Q$ is diagonal and by equality of the support $PQ$ is also diagonal. □

**Lemma 9** ($PQ = I_n$). *Let $Q$ be a well-partitioned lifting matrix. We have the following equivalence :*

$$PQ = I_n \iff P \in Q^g$$

*Proof.* Let's show the two sides of the equivalence.

$\implies$ If $PQ = I_n$, we have directly $\Pi^2 = \Pi$ and thus with Lem. 3, $Q \in P^g$. Then as $PQ = I_n$, we have $\text{rank}(P) = n$ and using Lem. 4 as $Q \in P^g$ we have $P \in Q^g$

$\impliedby$ if $P \in Q^g$ then $QPQ = Q$, by multiplying by the Moore Penrose inverse $Q^+$ to the left, we have $Q^+QPQ = Q^+Q$ thus $PQ = I_n$. The Moore Penrose inverse being a left inverse of $Q$ as seen in App. B.2 $\qquad\square$

## B  Proofs

### B.1  Consistency Lem. 2

This property is an extension of Loukas work [27] on the consistency of the combinatorial Laplacian. We will use the Lem. 1 proven in [27] for our proof.

Let $\mathcal{Q}$ be a well-partitioned lifting matrix. For all adjacency matrix $A$, consider $L(A) = \Delta(A)\mathcal{L}(A)\Delta(A)$ and $Q(A, \mathcal{Q}) = \Delta(A)^{-1}\mathcal{Q}\Delta(A_c) = \Delta^{-1}\mathcal{Q}\Delta_c$.

$\impliedby$ When $\mathcal{Q}$ is a binary matrix. By using Lem. 1 $\mathcal{L}(A_c) = \mathcal{Q}^\top\mathcal{L}(A)\mathcal{Q}$. Thus,

$$
\begin{aligned}
Q^\top LQ &= Q(A, \mathcal{Q})^\top L(A)Q(A, \mathcal{Q}) \\
&= \Delta_c\mathcal{Q}^\top\Delta^{-1}\Delta\mathcal{L}(A)\Delta^{-1}\mathcal{Q}\Delta_c \\
&= \Delta(A_c)\mathcal{Q}^\top\mathcal{L}(A)\mathcal{Q}\Delta(A_c) \\
&= \Delta(A_c)\mathcal{L}(A_c)\Delta(A_c) \\
&= L(A_c) \\
&= L_c
\end{aligned}
$$

$\implies$ When $L_c = Q^\top LQ$ then

$$
\begin{aligned}
L_c = Q^\top LQ &\implies \Delta_c\mathcal{L}_c\Delta_c = \Delta_c\mathcal{Q}^\top\Delta^{-1}\Delta\mathcal{L}\Delta\Delta^{-1}\mathcal{Q}\Delta_c \\
&\implies \Delta_c\mathcal{L}_c\Delta_c = \Delta_c\mathcal{Q}^\top\mathcal{L}\mathcal{Q}\Delta_c \\
&\implies \mathcal{L}_c = \mathcal{Q}^\top\mathcal{L}\mathcal{Q} \\
&\implies \mathcal{Q} \text{ is binary}
\end{aligned}
$$

The last line using Lem. 1.

### B.2  Proof of Lemma Moore Penrose inverse

*Proof.* Let's show that $Q^+ = (Q^\top Q)^{-1}Q^\top$.

For a given well-partitioned lifting matrix $Q \in \mathbb{R}^{N \times n}$, we have $\text{rank}(Q) = n$ as proposed in Lem. 7.

We can compute one left inverse as $Q_L^{-1} = (Q^\top Q)^{-1}Q^\top$.

It verifies the four properties of Moore Penrose inverse :
  1. $QQ_L^{-1}Q = Q(Q^\top Q)^{-1}Q^\top Q = Q$
  2. $Q_L^{-1}QQ_L^{-1} = (Q^\top Q)^{-1}Q^\top Q(Q^\top Q)^{-1}Q^\top = (Q^\top Q)^{-1}Q^\top = Q_L^{-1}$
  3. $(QQ_L^{-1})^\top = ((Q^\top Q)^{-1}Q^\top)^\top Q^\top = Q((Q^\top Q)^{-1})^\top Q^\top = QQ_L^{-1}$
  4. $(Q_L^{-1}Q)^\top = I_n = Q_L^{-1}Q$

Thus it is the unique Moore Penrose inverse of the lifting matrix $Q$.

Please note that for the third condition we used that $(Q^\top Q)$ diagonal (see Lem. 8) and so $(Q^\top Q)^{-1}$ is symmetric.

It is thus by definition a generalized reflexive inverse of $Q$ with the same support as $Q^\top$. $\qquad\square$

### B.3 Proof $P_{Loukas}$ generalized reflexive inverse

For a multilevel coarsening scheme we have intermediary coarsening with intermediary well-partitioned $Q_i$. We remark that $Q = Q_1 \ldots Q_c$ is also a well-partitioned lifting matrix.

Let's examine $P_{Loukas} = Q_c^+ \ldots Q_1^+$: First it is of same support as $Q^\top$. Secondly, $P_{Loukas}Q = Q_c^+ \ldots Q_1^+ Q_1 \ldots Q_c = I_n$ as Moore Penrose inverse are left inverse. Using Lem. 9, we have $P \in Q^g$. Thus $P_{Loukas}$ is a generalized reflexive inverse of $Q$ and has the same support as $Q^\top$.

### B.4 Proof of Coarsen-lift operator projection Lem. 3

*Proof.* Let's show the two side of the equivalence :

$\implies \Pi$ **projection implies** $Q \in P^g$ :

We have $\Pi^2 = \Pi$ then $QPQP = QP$.

As $Q$ is well-partitioned, we have that $\text{rank}(Q) = n$ (using Lem. 7). Thus we have the existence of a left inverse ( one example is the Moore Penrose inverse) such that $Q_L^{-1}Q = I_n$.

We multiply our expression at the left by $Q_L^{-1}$, then $Q_L^{-1}QPQP = Q_L^{-1}QP$ thus $PQP = P$. Consequently $Q$ is a generalized inverse of $P$.

$\impliedby Q \in P^g$ **implies** $\Pi = QP$ **projection** :

$\Pi\Pi = QPQP = QP = \Pi$. Thus $\Pi$ is a projection using directly the generalized inverse property on $Q$. (condition (i) ). $\qquad \square$

### B.5 Proof of Reflexive generalized inverse Lem. 4

*Proof.* For a well-partitioned lifting matrix $Q$ and a reduction matrix $P$ such that $Q \in P^g$. Let's show the two sides of the equivalence:

$\implies \text{rank}(P) = n$ **implies** $P \in Q^g$ :

With this "full rank" reduction matrix, using Lem. 7 there is the existence of a right pseudo inverse $P_R^{-1}$ such that $PP_R^{-1} = I_n$.

As we have $Q \in P^g$ that implies $\Pi^2 = \Pi$, using the Lem. 3. We thus have $QPQP = QP$ using the existence of $P_R^{-1}$ we multiply this equality by $P_R^{-1}$.

$$QPQP = QP \implies QPQPP_R^{-1} = QPP_R^{-1}$$
$$\implies QPQ = Q$$

Thus $P \in Q^g$.

$\impliedby P \in Q^g$ **implies** $\text{rank}(P) = n$ :

$QPQ = Q$ by assumption. As we know that $\text{rank}(Q) = n$, we can compute one left inverse $Q_L^{-1}$ and multiply at left for both side of this equation. Thus we have $PQ = I_n$.

This is only possible if $\text{rank}(P) \geq n$ otherwise the kernel of $PQ$ would not be null. Thus bounded by the dimension of the matrix $\text{rank}(P) = n$.

**Second implication of the theorem** Now for the second part of the theorem: we have $P \in Q^g$. Thus $PQPQ = PQ$ and $\Pi^2 = \Pi$.

Using the Lem. 3, we have $Q \in P^g$. this justifies the inclusion of $E_2 \subset E_1$ $\qquad \square$

### B.6 Proof of Generalized inverse Characterization Lem. 5

*Proof.* We will use the following theorem presented as theorem 2.1 in [32], but presented as a Corollary of [31] in [2].

**Theorem 1** (Generalized inverse characterization). *Let $A \in \mathbb{R}^{m \times n}$. Then $A^g$ exists. The entire classe of generalized inverses is generated from any given inverse $A^g$ by the formula*

$$A^g + U - A^g A U A A^g \tag{16}$$

*where $U \in \mathbb{R}^{n \times m}$ is arbitrary.*

We apply this characterization to the well-partitioned lifting matrix $Q \in \mathbb{R}^{N \times n}$ using a well known generalized inverse of $Q$, namely the "Moore-Penrose pseudo inverse" $Q^+$ that has been characterized in App. B.2.

Thus $Q^+ + M - Q^+ Q M Q Q^+$, for $M \in \mathbb{R}^{n \times N}$ arbitrary, generates all the generalized inverse of the lifting matrix $Q$.

But this formula has some simplifications. Indeed $Q^+ Q = I_n$ as we have proved in App. B.2. Thus the characterization can be rewritten as $Q^+ + M(I_n - Q Q^+)$ for an arbitrary $M \in \mathbb{R}^{n \times N}$.

**Reflexive**   Let's show that these generalized inverse are also reflexive :

We have $QPQ = Q$ as $P \in Q^g$. Thus $\Pi^2 = QPQP = QP = \Pi$. Using the equivalence of Lem. 3 we have that $Q \in P^g$. This is a characterization of reflexive generalized inverse.

$\square$

### B.7   Proof of Generalized reflexive inverse of same support Lem. 6

*Proof.* Let's show the two side of the equivalence :

$\Longrightarrow P \in Q^g$ **with same support implies** $\sum_{k=1}^{N} \frac{P_{ik}}{\Delta(k)} = \frac{1}{\Delta_c(i)}$    :

For a well-partitioned, degree-wised valued matrix $Q$ and a matrix $P \in Q^g$ with same support as $Q^\top$, using Lem. 9, we have $PQ = I_N$. Thus each diagonal term must be equal to one :

$$\forall i, \ (PQ)_{ii} = 1 \Longrightarrow \forall i, \ \sum_{k=1}^{N} P_{ik} Q_{ki} = 1$$

$$\Longrightarrow \forall i, \ \sum_{k=1}^{N} P_{ik} \frac{\Delta_c(i)}{\Delta(k)} = 1$$

$$\Longrightarrow \forall i, \ \sum_{k=1}^{N} \frac{P_{ik}}{\Delta(k)} = \frac{1}{\Delta_c(i)}$$

It is the normalization condition we have.

$\Longleftarrow \sum_{k=1}^{N} \frac{P_{ik}}{\Delta(k)} = \frac{1}{\Delta_c(i)}$ **and with same support as** $Q^\top$ **implies** $P \in Q^g$   :

For a reduction matrix $P$ with the same support as $Q^\top$, $PQ$ is thus a diagonal matrix (Lem. 8).

Moreover $\sum_{k=1}^{N} \frac{P_{ik}}{\Delta(k)} = \frac{1}{\Delta_c(i)}$ we have

$$(PQ)_{ij} = \sum_{k=1}^{N} P_{ik} Q_{kj}$$

$$= \sum_{k | Q_{kj} \neq 0, Q_{ki} \neq 0} P_{ik} \frac{\Delta_c(i)}{\Delta(k)}$$

$$= \begin{cases} 0 \text{ when } i \neq j \\ 1 \text{ when} i = j \end{cases}$$

Thus $PQ = I_n$, using Lem. 9 we have $P \in Q^g$.

$\square$

## C  Presentation of datasets

We restrict the well known Cora and Citeseer to their principal connected component(PCC) as it more compatible with coarsening as a preprocessing step. Indeed, the loukas algorithm tend to coarsen first the smallest connected components before going to the biggest which leads to poor results for coarsening with a small coarsening ratio. However working with this reduced graph make the comparison with other works more difficult as it is not the same training and evaluating dataset. The characteristics of these datasets and their principal connected component are reported in Tab. 2.

Table 2: Characteristics of Cora and CiteSeer Datasets

| Dataset | # Nodes | # Edges | # Train Nodes | # Val Nodes | # Test Nodes |
|---|---|---|---|---|---|
| Cora | 2,708 | 10,556 | 140 | 500 | 1,000 |
| Cora PCC | 2,485 | 10,138 | 122 | 459 | 915 |
| Citeseer | 3,327 | 9,104 | 120 | 500 | 1,000 |
| Citeseer PCC | 2,120 | 7,358 | 80 | 328 | 663 |

## D  Random Geometric graphs

A random geometric graph is built by sampling nodes with coordinates in $[0, 1]^2$ and connecting them if their distance is under a given threshold. For this additional experiment on minimizing the RSA, we sample 1000 nodes with a threshold of $0.05$ (Fig. 3).

The results presented in Fig. 4 confirms the observation made for the same experiment on Cora, $P_{opt}$ being the best option to minimize the RSA.

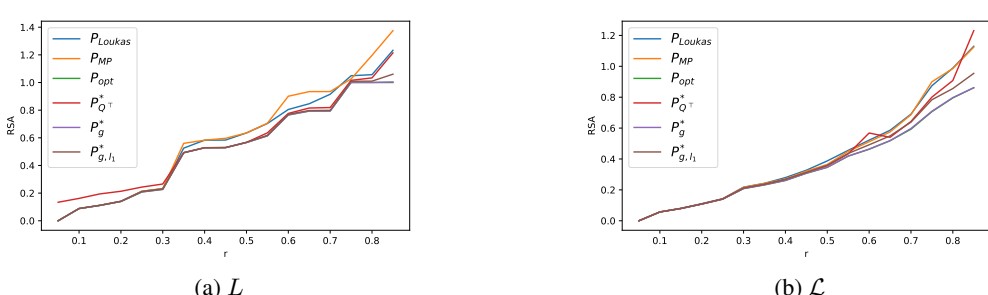

(a) $L$             (b) $\mathcal{L}$

Figure 4: RSA minimization on a Random Geometric Graph

## E  Convergence of optimization algorithm

For the optimization hyperparameters, we compared different optimizer, different learning rate, different initialization. The results for the three different problems can be find below.

### E.1  Parameters for $P_g^*$

We present here the chosen parameters for the optimization problem defined in Eq. 12.

As initialization matrix $M$ for our optimization procedure, we choose the Moore Penrose inverse matrix $P_{MP}$ compared to $P_{opt}$ and $P_{Loukas}$. Indeed as shown in Fig. 5, the $P_{opt}$ initialization is too

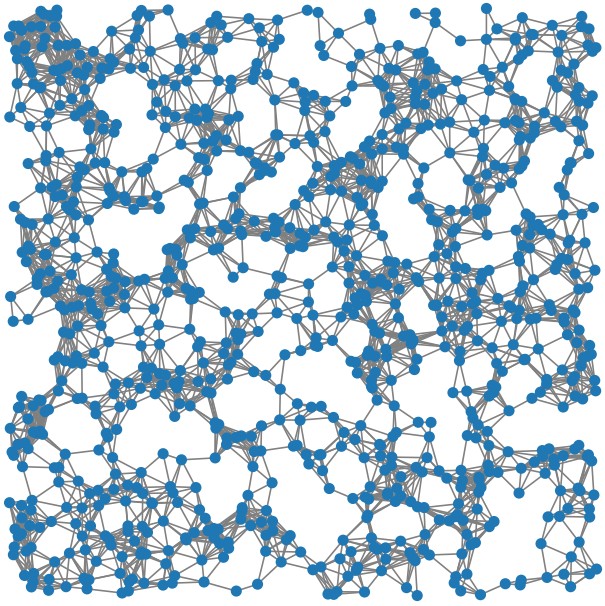

Figure 3: Example of a random Geometric graph

close to the minima and thus the matrix doesn't change and is too similar to $P_{opt}$. $P_{MP}$ is a better initialization for combinatorial Laplacian as $P_{Loukas}$. Furthermore $P_{MP}$ is more generic as $P_{Loukas}$ which can only be computed in a multi-level coarsening algorithm. The random initialization does not converge fast enough and show the relevance to consider a more "classic" reduction matrix as initialization.

We choose the stochastic gradient optimizer (SGD) as it is more stable than the Adam optimizer. For the Learning rate we choose $lr = 0.01$ for an improved stability.

Fig. 5 is computed for a coarsening on Cora with $r = 0.5$.

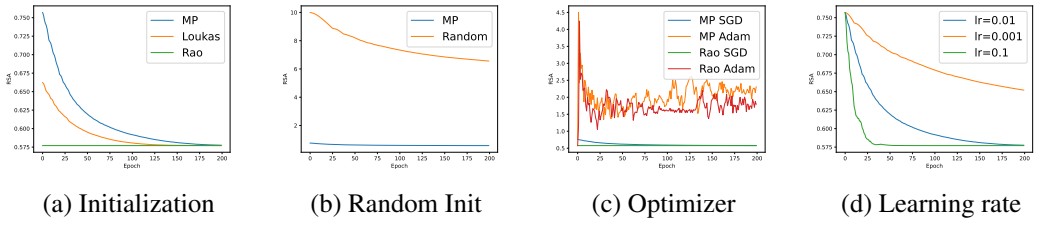

(a) Initialization      (b) Random Init      (c) Optimizer      (d) Learning rate

Figure 5: Parameters for $P_g^*$

## E.2   Parameters for $P_{g,l_1}^*$

We present here the chosen parameters for the optimization problem defined in Eq. 13.

To enforce the sparsity we apply after the optimization procedure a threshold of $0.001$ to erase the small coefficients.

We choose as initialization matrix $M$ the $P_{opt}$ matrix. $P_{MP}$ and $P_{Loukas}$ provide a better combined loss as shown in Fig. 6, but due to the sparsity constraint they do not leave their support and the

sparsity remains unchanged (as shown in Tab. 3). It then becomes an optimization on the same support and the RSA is less improved. Oppositely when we initialize with $P_{r}ao$ which is very dense, we obtain a number of non zero coefficients close to the support of $P_{MP}$ with an additional 500 coefficients.

We choose an $\lambda$ coefficients which control the $l_1$ penalty equals to $\lambda = 0.01$ to enforce the sparsity.

As the previous experiment, we choose the SGD optimizer and a learning rate $lr = 0.01$ for an improved stability.

Fig. 6 and Tab. 3 are computed for a coarsening on Cora with $r = 0.5$.

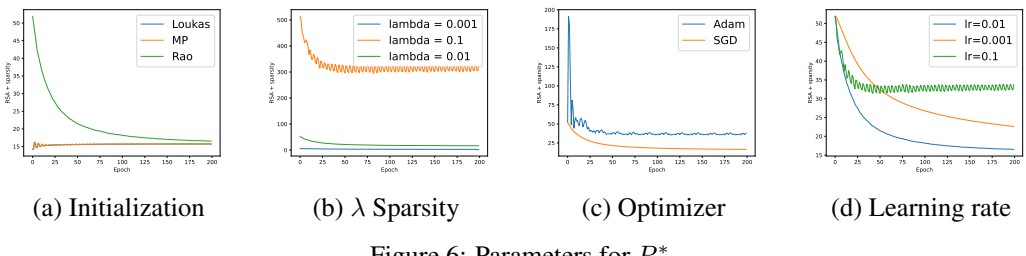

(a) Initialization     (b) $\lambda$ Sparsity     (c) Optimizer     (d) Learning rate

Figure 6: Parameters for $P_g^*$

Table 3: Influence of Sparsity coefficient $\lambda$ for $P_{g,l_1}^*$ (ex with $r = 0.5$ and threshold $= 0.001$)

| Initialization | $\lambda$ | #Non zero coefficients of Initialization | # Non zero Coefficients |
|---|---|---|---|
| $P_{Loukas}$ | 0.01 | 2,485 | 2,479 |
| $P_{MP}$ | 0.01 | 2,485 | 2,509 |
| $P_{opt}$ | 0.01 | 3,080,144 | 3,007 |
| $P_{opt}$ | 0.001 | 3,080,144 | 29,585 |
| $P_{opt}$ | 0.1 | 3,080,144 | 654,080 |

## E.3   Parameters for $P_{Q^\top}^*$

We present here the chosen parameters for the optimization problem defined in Eq. 14.

As initialization vector $\mu$ for our optimization procedure, we choose the non zero coefficients of the Moore Penrose inverse matrix $P_{MP}$ compared to the non zero coefficient of $P_{Loukas}$, a random initialization and the uniform vector which has all same values for each node in the same super node. Indeed as shown in Fig. 7, the $P_{MP}$ vector initialization is the only stable method.

The SGD optimizer is still more stable than the Adam optimizer which motivate its choice. For the Learning rate we choose $lr = 0.05$ for a fast convergence and an improved stability. The results are reported in Fig. 7

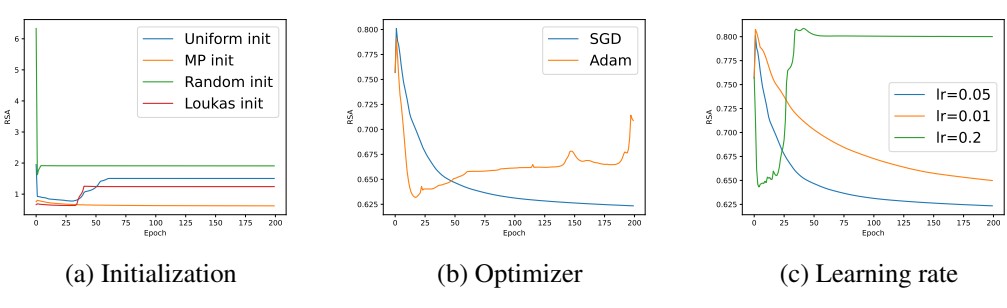

(a) Initialization     (b) Optimizer     (c) Learning rate

Figure 7: Parameters for $P_{Q^\top}^*$

Fig. 7 is computed for a coarsening on Cora with $r = 0.5$.

# F   Adaptation of Loukas Algorithm

You can find below the pseudo-code of Loukas algorithm Alg. 1. This algorithm works by greedy selection of *contraction sets* (see below) according to some cost, merging the corresponding nodes, and iterate. The main modification is to replace the combinatorial Laplacian in the Loukas code by any Laplacian $L = \Delta \mathcal{L} \Delta$. Note that we also remove the diagonal of $A_c$ at each iteration, as Loukas given its lower value of RSA. The output of the algorithm is the resulting lifting matrix $Q = Q_1 \ldots Q_c$, the coarsened adjacency $A_c$ and the the Loukas and Moore Penrose reduction matrices $P_{Loukas}$ and $P_{MP}$.

---

**Algorithm 1** Loukas Algorithm

---

**Require:** Adjacency matrix $A$, Laplacian $L = \Delta \mathcal{L} \Delta$, a coarsening ratio $r$ , preserved space $\mathcal{R}$, percentage number of nodes merged at one coarsening step : $n_e$
1:  $n_{obj} \leftarrow \text{int}(N - N \times r)$ the number of nodes wanted at the end of the algorithm.
2:  compute cost matrix $B_0 \leftarrow VV^\top L^{-1/2}$ with $V$ an orthonormal basis of $\mathcal{R}$
3:  $Q \leftarrow I_N$
4:  **while** $n \geq n_{obj}$ **do**
5:      Make one coarsening STEP $l$
6:      Create candidate contraction sets.
7:      For each contraction $\mathcal{C}$, compute $\text{cost}(\mathcal{C}, B_{l-1}, L_{l-1}) = \frac{\|\Pi_C B_{l-1}(B_{l-1}^\top L l-1 B_{l-1})^{-1/2}\|_{\mathcal{L}_\mathcal{C}}}{|\mathcal{C}|-1}$
8:      Sort the list of contraction set by the lowest score
9:      Select the lowest scores non overlapping contraction set while the number of nodes merged is inferior to $\min(n - n_{obj}, n_e)$
10:     Compute the binary 0-1 matrix $\mathcal{Q}_l$ intermediary lifting matrix with contraction sets selected
11:     $Q_{l-1}Q_l \leftarrow \Delta_{l-1}^{-1} \mathcal{Q}_l \Delta_l$
12:     $P_{Loukas} \leftarrow Q_l^+ P_{Loukas}$
13:     $P_{MP} \leftarrow \text{reuniform}(Q_l^\top) P_{MP}$
14:     $B_l \leftarrow \mathcal{Q}^+ B_{l-1}$
15:     $A_l \leftarrow \mathcal{Q}^\top A_{l-1} \mathcal{Q} - \text{diag}\left( (\mathcal{Q}^\top A_{l-1} \mathcal{Q}) 1_n \right)$
16:     $L_l \leftarrow Q^\top L_{l-1} Q$
17:     $n \leftarrow \min(n - n_{obj}, n_e)$
18: **end while**
19: $P_{MP} \leftarrow (Q^\top Q)^{-1} Q^\top$
20: **return** $A_c, Q, P_{Loukas}, P_{MP}$

---

The terms $\Pi_\mathcal{C}$ and $L_\mathcal{C}$ denote specific projection of the contraction set. Their precise definitions are provided in Loukas work [27]. We kept them unchanged in our experiments and defer any potential adjustments to future work.

In our adaptation we also add a parameter $n_e$ to limit the number of nodes contracted at each coarsening step. In one coarsening step, when a contraction set $\mathcal{C}$ is selected, we merge $|\mathcal{C}|$ nodes. In practice Loukas proposed in its implementation to force $n_e = \infty$ and coarsen the graph in one single iteration. We observed empirically that decreasing $n_e$ leads to improved results.

**Candidate contraction Set.**   Candidate contractions sets can take two main forms: either pairs of nodes connected by an edge (referred to as the variation edges version), or the full neighborhood of each nodes (the variation neighborhood version). In practice, since neighborhoods tend to be large in our graph, this second option proves impractical for small coarsening ratios and typically leads to suboptimal results. We therefore rely on edge-based candidate sets, and adjust the parameter $n_e$ to control the greedy behaviour of the algorithm.

# G   Experiments hyperparameters

For all the experiments, we preserve $K$ eigenvectors of the normalized self looped Laplacian $L = \Delta \mathcal{L} \Delta$ with $\Delta = (\operatorname{diag}(A1_N) + 1)^{-1/2}$ with $K = 100$. We apply our adapted version of Loukas variation edges coarsening algorithm with $n_e = 10\% N$.

For the optimization hyperparameters as discussed previously in the appendix, we choose :
- $P_{Q^\top}^*$ is initialized with the non zero coefficients of $P_{MP}$. We use a SGD optimizer with a momentum of 0.9, a learning rate $lr = 0.05$ and 200 epochs.
- $P_g^*$ is initialized with the non zero coefficients of $P_{MP}$. We use a SGD optimizer with a momentum of 0.9, a learning rate $lr = 0.01$ and 200 epochs.
- $P_{g,l_1}^*$ is initialized with the non zero coefficients of $P_{opt}$ and we use a $l_1$ penalty coefficient $\lambda = 0.01$ and a threshold of 0.001 to enforce the sparsity. We use a SGD optimizer with a momentum of 0.9, a learning rate $lr = 0.01$ and 200 epochs.

## G.1   Hyperparameters for Tab. 1

For the GCN, for both Cora and CiteSeeer we have 3 convolutional layers with the hidden dimensions $[256, 128]$. We use an Adam Optimizer with a learning rate $lr = 0.01$ and a weight decay $wd = 0.001$.

For the SGC model on Cora and Citeseer we make 2 propagations as preprocessing for the features. We use an Adam Optimizer with a learning rate $lr = 0.1$ and a weight decay $wd = 0.001$.

## G.2   Sparsity of the reduction matrices for Tab. 1

Table 4: Number of non zero elements for the different reduction matrices of Tab. 1

| #Non zero coefficients | Cora | | | Citeseer | | |
|---|---|---|---|---|---|---|
| $r$ | 0.3 | 0.5 | 0.7 | 0.3 | 0.5 | 0.7 |
| $P_{Loukas}$ | 2,485 | 2,485 | 2,485 | 2,120 | 2,120 | 2,120 |
| $P_{MP}$ | 2,485 | 2,485 | 2,485 | 2,120 | 2,120 | 2,120 |
| $P_{opt}$ | 4,315,200 | 3,080,144 | 1,846,584 | 3,136,320 | 2,235,527 | 1,341,714 |
| $P_{Q^\top}^*$ | 2,485 | 2,485 | 2,485 | 2,120 | 2,120 | 2,120 |
| $P_g^*$ | 2,261,446 | 2,202,595 | 1,596,687 | 1,505,412 | 1,533,820 | 1,131,521 |
| $P_{g,l_1}^*$ | 2,822 | 3,003 | 2,881 | 2,629 | 3,505 | 3,278 |
| $P_{opt}$ (coeff > 0.001) | 583,431 | 766,439 | 580,971 | 537,353 | 594,807 | 421,027 |
| $P_g^*$ (coeff > 0.001) | 5,735 | 46,080 | 99,288 | 18,601 | 24,874 | 79,609 |

