# OpenReview forum: "Taxonomy of reduction matrices for Graph Coarsening"
_NeurIPS.cc/2025/Conference — NeurIPS 2025 poster_

### Official Review · Reviewer_mmpw · 2025-06-25

**Clarity:** 4
**Significance:** 3
**Originality:** 3
**Rating:** 5
**Confidence:** 4

**Summary:**

Graph coarsening is an important algorithmic primitive for many pipelines involving graphs in ML, such as graph/geometric deep learning, but also computational methods working on or with graphs beyond ML. There are celebrated results on spectral approximations of large graphs by sparser ones (see the work of Spielman for instance) but fewer strong theoretical results on reducing/coarsening large graphs to smaller ones with mathematical guarantees.

In ML, it is of particular importance to preserve features defined over the graph and that can represent external signals or graph structure. This paper builds on and extends previous work by Lukas, widening the coarsening and lifting operations that lead to preserved spectral properties (defined via a stable embedding condition). A systematic exploration of the "design space" of these operations allow the authors to propose several new operations and recover older ones.

Experimental validation suggest that these techniques are particularly suited for homophilic/smooth signals, but less so when this restriction is lifted.

**Questions:**

- Please clarify the role of graph structure (beyond the example of the random geometric graph) and graph signal models (is the method suited for non-smooth graph signals at all)
- Given the importance of Kron reduction in many fields (it has been used in graph signal processing/ML), and its well-studied spectral properties, I think it deserves a more elaborate discussion: can you compare both approaches, what would be the advantages/disadvantages of both. Kron reduction can also be (at least partially) inverted using its spectral approximation properties, is there any link to what is being proposed in this paper ?

**Ethical Concerns:**

["NO or VERY MINOR ethics concerns only"]

**Final Justification:**

Having interacted with the authors and read their rebutal, I think the paper has gained in intelligibility and impact. I have therefore updated ymy score accordingly

**Limitations:**

Yes, except maybe on the practical limitations of the methods. I do not consider this critical though.

**Paper Formatting Concerns:**

I only would like to thank the authors for the great care they showed in preparing this manuscript

**Quality:**

3

**Strengths And Weaknesses:**

Strengths:
- very clearly written, the paper is easy to follow with crystal motivations
- strong mathematical results, even if limited by the spectral objectives (which one must say are the norm in this field)

Weaknesses:
- how important is the structure of the graph ?
- it is hard to tell whether the theoretical motivation results in practical gains
- practical results are limited to rather small graphs where one could question the usefulness of the whole endeavour

---

> ### Author Rebuttal · Authors · 2025-07-30
>
> Thank you for your review and positive comment on the clarity of the motivation. We address your remarks below.
>
> **W1, Q1)** *Please clarify the role of graph structure (beyond the example of the random geometric graph) and graph signal models (is the method suited for non-smooth graph signals at all)*
>
> We chose to focus on the RSA score as it is a standard spectral objective in graph coarsening. However, our optimization procedure is compatible with any scoring function, allowing future extensions to alternative objectives.
> The role of the graph structure in our approach relies heavily on the Laplacian and its low-frequency components, as shown in the first lemma. The RSA is designed to estimate the reconstruction of smooth (low-frequency) signals within a preserved subspace $\mathcal{R}$. Exploring alternatives scoring functions designed for different types of signals, including non-smooth ones is a promising direction but remains out of scope for this paper.
>
> **W2)** *it is hard to tell whether the theoretical motivation results in practical gains*
>
> This works effectively shows that the RSA could be optimized through the reduction matrices in different subsets. However, the connection between RSA and GNN performance, proved in a prior work [22] (with a theoretical upper bound that may be quite loose), is sometimes not perfectly reflected in our results, as other phenomena probably comes into play. This suggests that future work should explore alternative scoring functions, or even consider directly optimizing the coarsening in a supervised manner. This point will be added to the discussion on future work in the final version.
>
> **W3)** *practical results are limited to rather small graphs where one could question the usefulness of the whole endeavour*
>
> In our setting, RSA is made differentiable, which is more costly than the non-differentiable objectives typically used with sparse methods. This limits scalability to large graphs like OGB-Arxiv or Reddit. Designing scalable differentiable scoring functions is a key direction for future work. This point will be clarified in the final version of the paper.
>
> **Q2)** *Given the importance of Kron reduction in many fields (it has been used in graph signal processing/ML), and its well-studied spectral properties, I think it deserves a more elaborate discussion: can you compare both approaches, what would be the advantages/disadvantages of both. Kron reduction can also be (at least partially) inverted using its spectral approximation properties, is there any link to what is being proposed in this paper?*
>
> Kron reduction plays a central role in many areas of graph machine learning. In the context of graph coarsening, it can be seen as an alternative to Loukas' coarsening algorithm. Since this algorithm was specifically designed to minimize the RSA, we focus our experiments on it. However, our optimization scheme is compatible with any scoring function or coarsening method. The coarsening algorithm is only used to define the contraction sets and thus the lifting matrix $Q$, before optimizing the reduction matrix $P$. We will clarify in the final version that the proposed optimization method applies to any coarsening algorithm.

---

> > ### Comment · Reviewer_mmpw · 2025-08-01
> >
> > Thank you for your detailed answers. Given that your scheme can handle other coarsening methods, irrespective of the RSA, I am curious how the *practical* results would change on any of your experiments, for instance using Kron instead of Loukas. Is it something doable within the timeframe of this rebuttal ?

---

> > > ### Author Response · Authors · 2025-08-04
> > >
> > > Thank you for the suggestion. We will do our best, but given that Kron is not initially designed for the RSA, it may be a bit difficult to calibrate the experiments within the time frame of the rebuttal. In which case, we will do our best to include some experiments in this direction in the final version of the paper.

---

### Official Review · Reviewer_QiV5 · 2025-06-27

**Clarity:** 3
**Significance:** 3
**Originality:** 3
**Rating:** 5
**Confidence:** 4

**Summary:**

This paper studies the admission space of reduction matrices and its optimization towards preserving Restricted Spectral Approximation (RSA) in graph coarsening. Specifically, the authors observe that the coarsening can be solely defined by the lifting matrix and then decouple the relation between the lifting and reduction matrix, which is typically chosen as pseudo-inverse of each other. With this, they reveal three admission spaces of the reduction matrix, from projection to generalized reflexive inverse and to generalized reflexive inverse of the same support. They then study optimizing the reduction matrix for the RSA objective and discuss six reduction matrices under different admission spaces and sparsity constraints. Experiments have been conducted on two small datasets to demonstrate the RSA preservation for different reduction matrices and their impacts on the accuracy of the GNN application.

**Questions:**

Comments:

The lifting and reduction matrices are often considered to be pseudo-inverse of each other with the intuition that the two processes are inverse. This paper defines the adjacency matrix of the coarsened graph as (1) and optimizes P. Would it be possible to use P to define the coarsened graph and then optimize Q for the RSA target instead?

The well-partitioned property may not be always true in real-life coarsening tasks with possibly overlapping clusters/supernodes. What are known results for the overlapping setting?

The computation overhead for getting different reduction matrices should be provided to offer guidance in the selection.

I’m quite curious on the empirical performance of GNN on the RSA approximation and data-driven graph pooling methods [39,13,34]. RSA may have better generalization on different downstream tasks while data-drive methods are optimized for specific task and data.

Typo: Line 135: a eigen-subspace.

Line 43: “In virtually all works on graph coarsening, these two matrices are simply pseudo-inverse of each other, and both can represent the “graph coarsening” indifferently.” and Line 85: “These methods use interchangeably the reduction and the lifting matrix by choosing one as the transposed of the other.” Graph coarsening and pooling seem to be similar, if not the same. Why does one set the two matrices to be pseudo-inverse of each other and the other set them to be transposed of each other? Please clarify.

**Ethical Concerns:**

["NO or VERY MINOR ethics concerns only"]

**Final Justification:**

The authors answered well in the rebuttal and I am keeping my score.

**Limitations:**

I did not see the discussion but this work may not have clear negative societal impacts that have to be explained. Should add limitation discussions though.

**Paper Formatting Concerns:**

No issues identified.

**Quality:**

3

**Strengths And Weaknesses:**

Strengths:

The paper establishes a good understanding between the relations between the lifting and reduction matrices and pointed out the lifting matrix defines the coarsening while the reduction matrix can affect the RSA preservation and thus can be optimized towards a better preservation. It is a solid paper that builds on top of the RSA objective and can be of great reference for similar research in graph coarsening.

Analyze the three levels of admission spaces for the reduction matrix and formulate the optimization over the admission spaces. Discuss the optimization and special properties of each reduction matrix, ranging from typical Moore-Penrose inverse to RSA optimization with a sparsity constraint. Although I did not review the proof in Appendix, the conclusions look natural and should be correct.

While the main contribution is in theory, the authors have performed experiments to study the empirical performance on the RSA optimization and the impacts on downstream GNN application.

Weaknesses:

The spectral guarantee using RSA proposed in 2019 is only one type of key properties graph coarsening approximates. It is not clear how the developed optimization and coarsening framework have significant impacts on other types of graph coarsening objective.

The experiments were conducted on small-scale graphs, while this is a common problem for spectral methods. The empirical computation costs for different optimizations are missing and should be reported.

---

> ### Author Rebuttal · Authors · 2025-07-30
>
> Thank you for your review and feedback. We appreciate your positive assessment and have addressed your comments below.
>
> **W1)** *It is not clear how the developed optimization and coarsening framework have significant impacts on other types of graph coarsening objective.*
>
> Although we focus on optimizing the RSA score in this work, as it is a standard and interpretable objective in graph coarsening, the proposed framework is general and can be applied to other scoring functions, including other graph coarsening objectives. This flexibility enables broader applicability beyond RSA.
>
> **W2)** *The experiments were conducted on small-scale graphs, while this is a common problem for spectral methods. The empirical computation costs for different optimizations are missing and should be reported.*
>
> A discussion on the size of the optimized matrices and on large-scale graphs is already in the main paper to some extent, but we acknowledge that it could be extended. We briefly do so below. This discussion will be included in the final version of the paper.
>
> In $E_2$ at each optimization step we update the full matrix $P$ of size $n\times N$. This affects the memory complexity: $E_2$ is somewhat manageable for medium size graphs only. This is why we introduce $E_3$ with a support constraint, which is manageable for large graphs.
> Empirically, $E_2$ is highly parallelizable and the total runtime for the 200 iterations on a modern GPU last 5 seconds for Cora and Citeseer. $E_3$ is more memory efficient, but the renormalization tricks slightly slower it. Even on CPU, for Cora and Citeseer, the two optimization processes remain manageable and take less than 3 minutes.
>
> However, the main bottleneck of computational complexity lies in the computation of the scoring function : the differentiable RSA. Its computation requires the square root of the original Laplacian which is in $\mathcal{O}(N^3)$ and not feasible for large graphs such Reddit or OGB-Arxiv. While we limit this paper to the RSA because it is a classical criterion in graph coarsening that allows for interpretability as well as serves as inspiration for new reduction matrices, designing new, more scalable scoring function is a major path for future work.
>
> **Q1)** *Would it be possible to use P to define the coarsened graph and then optimize Q for the RSA target instead?*
>
> The asymmetry between $P$ and $Q$ is indeed the key point of the paper. In the classical definition of graph coarsening, it is $Q$ that is used to define the coarsened adjacency matrix $A_c$ and the coarsened Laplacian $L_c$ (which preserve the isometry with respect to $\lVert \cdot \rVert_{L_c}$), and thus $Q$ contains the structural information of the coarsening. In this framework, it therefore could not be easily optimized for the RSA target instead of $P$. Of course, one could imagine an entirely different framework for graph "coarsening" (since it is a vague problem at best) where constraints would lie elsewhere, however the framework that we use is the one largely admitted and employed in the literature. This will be clarified in the final version of this work.
>
> **Q2)** *The well-partitioned property may not be always true in real-life coarsening tasks with possibly overlapping clusters/supernodes. What are known results for the overlapping setting?*
>
> We appreciate the reviewer’s insightful suggestion. In the overlapping setting, it may be possible to relax the well-partitioned assumption by considering *a minima* a full-rank $Q$ matrix (i.e., $\operatorname{rank}(Q)= n $). In this case, the first two mathematical characterizations of $E_1$ and $E_2$ would still hold. However, computing the Moore–Penrose inverse becomes less convenient, as $Q^TQ$ is no longer diagonal and thus harder to invert. This nuance will be clarified in the final version of the paper. Future work could explore more intuitive ways to relax the well-partitioned assumption on the $Q$ matrix, beyond a simple rank-based argument: to remain interpretable and still represent a "coarsening", as opposed to a general reduction, the "overlapping" part should be quantified and limited. This represents an interesting mathematical challenge. We will add this discussion as outlooks.
>
> **Q3)** *The computation overhead for getting different reduction matrices should be provided to offer guidance in the selection.*
>
> Please refer to Weakness 2 answer.
>
> **Q4)** *I’m quite curious on the empirical performance of GNN on the RSA approximation and data-driven graph pooling methods [39,13,34].*
>
> We thank the reviewer for this interesting question. A possible future work indeed would be to consider this optimization process as part of the downstream task training. The GNN would be trained simultaneously with the reduction matrix $P$ and both the coarsened features $X_c = PX$ and the propagation scheme would be recomputed at each epoch $S_c = QSP$ with considering only one differentiable scoring function : the loss. This would slower the GNN training on the coarsened graph, and it could be hard to train as the number of hyperparameter would be important, but this idea seems promising for future works. We will include this remark in the outlook of the final paper.
>
> **Q5)** *Graph coarsening and pooling seem to be similar, if not the same. Why does one set the two matrices to be pseudo-inverse of each other and the other set them to be transposed of each other? Please clarify.*
>
> Graph coarsening and graph pooling are often likened in graph machine learning. Although similar, pooling is usually differentiable and used end-to-end to optimize GNN performance, but pays less attention to the lifting step (e.g. from a spectral point of view) and often uses transposes for computational efficiency. In contrast, coarsening is a *preprocessing* step designed to reduce memory and computation time for large graphs, and designed to promote information-preservation, often in an unsupervised manner. In the literature, it typically relies on pseudo-inverses for both lifting and reduction. It is worth noting that some pooling approaches also employ pseudo-inverses. So the two approaches may employ very similar tools, but often have slightly different objectives and methods. In our paper, we focus on unsupervised, spectral methods, which is closer to coarsening.

---

> > ### Comment · Reviewer_QiV5 · 2025-08-06
> >
> > Thank you for answering my questions. The end-to-end training in Q4 would provide empirical insights, considering that GNN application is already included in the work. For Q5, in addition to the computational efficiency, what are other differences between pseudo-inverses and transposes, e.g., are pseudo-inverses more aligned with spectral information preservation? In W1, it is still unknown how challenging the optimization after adapting to a new scoring function is.

---

> > > ### Author Response · Authors · 2025-08-06
> > >
> > > We thank the reviewer for their feedback and are happy to clarify the following points.
> > >
> > > While end-to-end training is indeed a promising direction, in this work the coarsening is considered as a task-independent preprocessing step. Moreover, integrating it into an end-to-end pipeline would require careful calibration (eg of many hyperparameters, for consistency across coarsening ratios), which is beyond the current scope.
> > >
> > > Regarding the second point, the transpose operators are not, in general, generalized inverses of each other. In particular, for a binary and well-partitioned matrix, this is not the case, and thus Lemma 3 no longer holds. As a result,the coarsen and then lift opererator $\Pi = QP$ is not a projection operator. This implies a weaker alignment with spectral preservation, as iteratively coarsening and lifting a signal may lead to further degradation.
> > >
> > > Finally we agree with the reviewer for W1, the complexity and stability of the optimization process depends on the chosen scoring function. Nevertheless, the sparsity of the reduction matrix provides a universal lever to control complexity and align with diverse objectives. It is very likely that optimizing over $E_3$ results in a more scalable optimization problem than $E_2$, regardless of the scoring function.

---

### Official Review · Reviewer_DYWE · 2025-07-01

**Clarity:** 3
**Significance:** 2
**Originality:** 2
**Rating:** 4
**Confidence:** 3

**Summary:**

This paper introduces a taxonomy of admissible families of reduction matrices for graph coarsening. The authors first extends the graph coarsening metric Restricted Spectral Approximation (RSA) by Loukas, in order to disentangle the lifting matrix and the reduction matrix. By observing the additional flexibility in defining reduction matrices, the authors proposes three nested sets of reduction matrices to optimize RSA. The authors provide concrete examples of these reduction matrix characterizations via pointers to the existing literature, and propose new choices of reduction matrices. These reduction matrices are compared empirically on the RSA metric and predictive performance of GNN trained on the obtained coarsened graphs.

**Questions:**

1. For the sparse reduction matrices in set $E_3$, the proposed $P^*_{Q^{\top}}$ empirically perform quite similar (within standard error bars) to the existing baselines $P_{Loukas}, P_{MP}$. Can the authors discuss their relationship in more details, either conceptually or empirically?

2. The authors discuss in the introduction that "$P$ is taken as the Moore-Penrose pseudo-inverse...In this paper, we challenge this choice and argue that there is no real reason for it.". But the characterization of $E_3$ clearly shows the advantages of $P$ being Moree-Penrose pseudo-inverse: (1) it is sparse (share the same support as $Q$); (2) it admits a close-form given $Q$, so no further optimization is needed. From the RSA evaluation, the pseudo-inverse $P$ choice underperforms $P^*_{Q^{\top}}$ requiring optimization. Do the authors agree? If so, I would suggest reword this discussion.

3. Although the characterization of $E_1, E_2$ reduction matrices is interesting, they introduce heavy computational overhead (Appendix G.2) and seem to bring very little advantage to GNN performance (Table 1). Can the authors discuss possible sparser characterizations or more efficient implementations?

Minor:
a) Definition 2 is the extended version of RSA, which differs from the one proposed by Loukas in disentangling the role of the lifting matrix and the reduction matrix. I would suggest adding the remark before/after this to remind the reader.

b) The notation $\Delta$ in Lemma 6: does it refer to the $\Delta$-Laplacian? I didn't find it defining anywhere close to Lemma 6 nor its proof.

**Ethical Concerns:**

["NO or VERY MINOR ethics concerns only"]

**Final Justification:**

The authors have addressed my main concern on complexity and applicability of the method (e.g., scalability, heterophilous graphs), and I have raised my score from 3 to 4. The complexity/scalability aspect is also brought up in other reviews (e.g., QiV5, btns), and the authors responded by highlighting the bottleneck $O(N^3)$ in the Laplacian computation for optimizing the chosen metric RSA. The authors acknowledged that considering other more computationally efficient metric is an interesting future direction but out of the scope of the current work. Considering this limitation, I thus recommend a weak acceptance of the paper.

**Limitations:**

yes

**Quality:**

3

**Strengths And Weaknesses:**

Strengths:
1. The proposal of a taxonomy of reduction matrices for graph coarsening is interesting and of great interest to the graph learning community.

2. The paper is well written overall, with a clear exposition of the motivation and chosen sets of reduction matrices.

Weakness:
1. Graph coarsening aims to reduce computation complexity (while maintaining performance). However, for the proposed sets of reduction matrices $E_1, E_2$, they both consist of dense matrices $\mathbb R^{n \times N}$. This introduces a heavy memory and time complexity overhead, as shown in Appendix G.2 (e.g., these reduction matrices consists of millions of parameters to be optimized for a medium size graph Cora or Citeseer). The authors do not provide any analysis on complexity and discussion on runtime.

2. The empirical validation is only carried out on two Cora and Citeseer benchmark graphs (with additional synthetic geometric random graphs). The paper can be further strengthened by a more thorough evaluation on more benchmark graphs including both homophilous ones and heterophilous ones (e.g., from [1] or [2], or even random graphs say a two-block stochastic block model where across-block connection probability is higher than within-block probability).

References
[1] Platonov, O., Kuznedelev, D., Diskin, M., Babenko, A., & Prokhorenkova, L. A critical look at the evaluation of GNNs under heterophily: Are we really making progress?. ICLR 2023
[2] Pei, H., Wei, B., Chang, K. C. C., Lei, Y., & Yang, B. Geom-GCN: Geometric Graph Convolutional Networks. ICLR 2020

---

> ### Author Rebuttal · Authors · 2025-07-30
>
> Thank you for your review and feedback. We are encouraged that you found the work of interest to the graph learning community. Please find our responses to your comments below.
>
> **W1)** *This introduces a heavy memory and time complexity overhead, as shown in Appendix G.2 (e.g., these reduction matrices consists of millions of parameters to be optimized for a medium size graph Cora or Citeseer). The authors do not provide any analysis on complexity and discussion on runtime.*
>
> As $E_1$ does not admit a closed-form expression, we focus our response on $E_2$ and $E_3$. Note that part of the discussion below is already present in the paper, we will complete it in the final version.
>
> In $E_2$, at each optimization step we update the full matrix $P$ of size $n\times N$. In $E_3$, we impose the support and thus we optimize the $N$ non-zero values of the matrix $P$. This affects the memory complexity: $E_2$ is only manageable for medium size graphs, while $E_3$ scales linearly in $N$. In terms of time complexity, $E_2$ is highly parallelizable and the total runtime for the 200 iterations on a modern GPU last 5 seconds for Cora and Citeseer. $E_3$ is more memory efficient, but the renormalization tricks make it slightly slower. Even on CPU, for Cora and Citeseer, the two optimization processes remain manageable and take less than 3 minutes, which is reasonable for a preprocessing done only once.
>
> Despite this, the main bottleneck that prevents scalability for now is in fact in the computation of the chosen scoring function: the differentiable RSA. It requires the square root of the original Laplacian which is in $\mathcal{O}(N^3)$. While we limit this paper to the RSA because it is a classical criterion in graph coarsening that allows for interpretability as well as serves as inspiration for new reduction matrices, designing new, more scalable scoring function is a major path for future work.
>
> **W2)** *The paper can be further strengthened by a more thorough evaluation on more benchmark graphs including both homophilous ones and heterophilous ones (e.g., from [1] or [2], or even random graphs say a two-block stochastic block model).*
>
> Similarly to the previous response, these limitations are mostly due to our choice to focus on the RSA, noting however that our optimization procedure is compatible with any scoring function, allowing future extensions to alternative objectives.
>
> Indeed, larger graphs such as OGB-Arxiv and Reddit were not included due to the high computational cost of optimizing a differentiable version of RSA in $\mathcal{O}(N^3)$.
>
> Similarly, heterophilous datasets are not really suitable for evaluating the RSA score, which is specifically designed to promote the preservation of low-frequency signals, by definition not compatible with heterophily. Indeed, an important direction for future work is to design scalable, heterophily-aware scoring functions. Exploring this idea in synthetic settings, such as a two-block stochastic block model, could serve as a natural first step.
>
> This point will be further emphasized in the discussion and limitations section of the final version.
>
> **Q1)** *For the sparse reduction matrices in set $E_3$, the proposed $P ^\ast_{Q^\top}$
> empirically perform quite similar (within standard error bars) to the existing baselines $P_{MP}$, $P_{Loukas}$ . Can the authors discuss their relationship in more details, either conceptually or empirically ?*
>
> As expected, the proposed $P ^\ast_{Q^\top}$ outperforms the naive baselines $P_{MP}$ and $P_{Loukas}$ *in minimizing the RSA score*. However, the connection between RSA and GNN performance proved in [22] is sometimes imperfect on real data, which is not really surprising as it is only an upper-bound and other criteria probably comes into play (sparsity of the message-passing, etc.). This suggests that future work should explore alternative scoring functions, or even consider directly optimizing the reduction matrix in a supervised manner, which we avoided here to keep the spirit of an unsupervised, general-purpose method not linked to any particular task. This discussion will be added in the final version.
>
> **Q2)** *But the characterization of $E_3$ clearly shows the advantages of being Moree-Penrose pseudo-inverse [...]*
>
> As the reviewer highlights, the Moore Penrose inverse is indeed a good choice in practice as it is sparse and has a close form given $Q$. But as shown in figure 2, $P ^\ast_{Q^\top}$ has a lower RSA. For the GNN application, since again the link between GNN performance and RSA is not perfect, the performance between these different choices can vary. This could change with another scoring function, as the proposed optimization process could work for any criterion.
>
> **Q3)** *Can the authors discuss possible sparser characterizations or more efficient implementations?*
>
> See our response to Weakness 1. Designing new scoring functions that scale to large graphs, by better leveraging the sparsity or allowing intermediate support sizes is a promising direction for future work, but remains out of scope for this paper.
>
> **Q4)** (minors): *Definition RSA and $\Delta$ notation*
>
> Thank you for the suggestion on the RSA definition. We agree that this addition will improve clarity, and we will make sure to include it in the final version of the paper.
>
> For the second point, $\Delta$ is indeed the diagonal matrix that it used for the normalization of the considered $\Delta$-Laplacian. We will make it clearer by adding an example for the combinatorial Laplacian in Lemma 6.

---

> > ### Comment · Reviewer_DYWE · 2025-08-04
> >
> > I thank the authors for their detailed response, which addressed my concern on the heterophilous dataset.
> >
> > Regarding the complexity: the authors pointed out that the major bottleneck lies in the $O(N^3)$ complexity in computing the square root of the Laplacian, for optimizing the differentiable RSA. However, the combinatorial Laplacian remains sparse, and can reduce the complexity to be linear with the number of edges. Does the $O(N^3)$ complexity bottleneck arise from computing the self-loop normalized Laplacian? In any case, it is worth pointing out explicitly where the $O(N^3)$ bottleneck occurs, and how one might avoid it (e.g., choosing a different Laplacian or a different score other than RSA).

---

> > > ### Author Response · Authors · 2025-08-05
> > >
> > > We thank the reviewer for their response and for engaging in the discussion.
> > >
> > > As stated in the main manuscript, the RSA involves the following explicit expression:
> > > $\lVert L^{1/2}(I_N − QP)V V^\top L^{-1/2} \rVert_2$.
> > >
> > > To compute this term, we require the eigendecomposition of the Laplacian $L= UMU^\top$  in order to construct both the square root and the inverse square root of $L$ (whether normalized or not). To our knowledge, this operation has a computational complexity of $\mathcal{O}(N^3)$ regardless of the sparsity of $L$.
> > >
> > > While there exist efficient iterative methods to approximate $L^{1/2}x$ and $L^{-1/2}x$ when applied to a vector $x$, these are not applicable here since we need the full matrix form in order to compute the spectral norm of the resulting matrix.
> > >
> > > Therefore, the main computational bottleneck does not come from computing the self-loop normalized Laplacian, but is inherent to the definition of the RSA. To overcome this limitation, one could consider using a more scalable score or fast approximate computation of the RSA, for instance based on random linear algebra. We will include this discussion in the final version of the paper.

---

> > > > ### Comment · Reviewer_DYWE · 2025-08-06
> > > >
> > > > Thank you for the clarification -- it addressed my concern on complexity. I will raise my score accordingly.

---

### Official Review · Reviewer_bTnS · 2025-07-05

**Clarity:** 3
**Significance:** 3
**Originality:** 3
**Rating:** 5
**Confidence:** 5

**Summary:**

This paper focuses on two detailed issues in graph coarsening: 1) What are the admissible degrees of freedom for the reduction matrix? and 2) Can we improve the RSA by simply modifying the reduction matrix alone? Overall, this paper conducts an in-depth and detailed theoretical study of graph coarsening operations and achieves valuable conclusions. I appreciate the work done in this paper.

**Questions:**

1. Please provide corresponding experiments to support the theoretical analysis. The specific experimental issues have already been mentioned in the weaknesses section.

**Ethical Concerns:**

["NO or VERY MINOR ethics concerns only"]

**Final Justification:**

After reviewing the other reviews, I have decided to raise the score to 5.

**Limitations:**

Yes

**Quality:**

3

**Strengths And Weaknesses:**

Strengths:

1.This paper is well-written and explains the background and context of the problem in detail.

2.The issues addressed in this paper are highly valuable for the field.

3.The theoretical analysis is solid.

 Weaknesses:

1.However, the experiments in this paper are still insufficient. Although this is a theoretical paper, I still hope the authors can provide more experimental results to support the theoretical conclusions. If the authors further expand the experiments, I will consider increasing my score. Specifically, I would like to see results on at least five datasets, including larger-scale datasets like Arxiv, which current graph coarsening algorithms are capable of handling.

---

> ### Author Rebuttal · Authors · 2025-07-30
>
> Thank you for your review and for engaging with our work. We are glad to hear you found the paper valuable for the field. Please find our response to your comment below.
>
> **Q 1)** *Please provide corresponding experiments to support the theoretical analysis*
>
> We acknowledge the concern regarding the number of datasets evaluated. In addition to the main experiments, we would like to point out that we already included results on random geometric graphs in the appendix for the RSA minimization.
>
> In this paper, we choose to focus our methodology on the RSA, a standard spectral objective in graph coarsening, since it allows for many interesting characterizations and intuitions pertaining to graph coarsening, as well as serves as inspiration for new reduction matrices. However, this comes with certain limitations, foremost among them the computational cost of this criterion. In particular, in our setting, we implement the RSA in a differentiable manner, which is more costly than the non-differentiable implementations that typically leverage sparse representations. This limits scalability to large graphs like OGB-Arxiv or Reddit. However, our optimization procedure is compatible with any scoring function, and designing scalable differentiable alternatives is a key direction for future work. This point will be clarified in the final version of the paper.

---

> > ### Comment · Reviewer_bTnS · 2025-08-08
> >
> > Thank you for your response. After reviewing the other reviews, I have decided to raise the score to 5.

---

### Decision · Program_Chairs · 2025-09-17

**Decision:**

Accept (poster)

**Comment:**

This paper introduces a taxonomy of admissible families of reduction matrices for graph coarsening. The authors first extend the graph coarsening metric Restricted Spectral Approximation (RSA) by Loukas, in order to disentangle the lifting matrix and the reduction matrix. By observing the additional flexibility in defining reduction matrices, the authors propose three nested sets of reduction matrices to optimize RSA. The authors provide concrete examples of these reduction matrix characterizations via pointers to the existing literature, and propose new choices of reduction matrices. These reduction matrices are compared empirically on the RSA metric and predictive performance of GNN trained on the obtained coarsened graphs.

The reviewers were positive about this paper, specifically stating that the topic is of interest to the general GNN community, and that the paper is clear and well written.
As indicated by the reviewers, scalability is a main limitation of the approach studied in this paper, as the procedure is cubic in the number of vertices.